# Teacher-Guided Student Self-Knowledge Distillation Using Diffusion Model

## Abstract

Existing Knowledge Distillation (KD) methods often align feature information between teacher and student by exploring meaningful feature processing and loss functions. However, due to the difference in feature distributions between the teacher and student, the student model may learn incompatible information from the teacher. To address this problem, we propose teacher-guided student **D**iffusion **S**elf-**KD**, dubbed as DSKD. Instead of the direct teacher-student alignment, we leverage the teacher classifier to guide the sampling process of denoising student features through a light-weight diffusion model. We then propose a novel locality-sensitive hashing (LSH)-guided feature distillation method between the original and denoised student features. The denoised student features encapsulate teacher knowledge and could be regarded as a teacher role. In this way, our DSKD method could eliminate discrepancies in mapping manners and feature distributions between the teacher and student, while learning meaningful knowledge from the teacher. Experiments on visual recognition tasks demonstrate that DSKD significantly outperforms existing KD methods across various models and datasets. Our code is attached in supplementary material.

## 1. Introduction

In the era of deep learning, how to compress large models while maintaining their performance has become a valuable topic. Knowledge Distillation (KD) is a well-known technique to achieve this goal. The core idea of KD is to transfer the knowledge from a pre-trained, high-capacity teacher network to a lightweight student network. With additional teacher guidance, the student performance could be significantly improved compared to training independently.

[1]Anonymous Institution, Anonymous City, Anonymous Region, Anonymous Country. Correspondence to: Anonymous Author <anon.email@domain.com>.

Preliminary work. Under review by the International Conference on Machine Learning (ICML). Do not distribute.

The original KD (Hinton et al., 2015) guides the student to learn from the teacher's final class probability predictions by KL divergence. This intuitive idea helps the student mimic the teacher's better outputs effectively. However, the final predictions lack intermediate information, making the student unable to understand the teacher's thinking process. Therefore, many works explore feature distillation methods that distill intermediate feature maps (Romero et al., 2014) or their extracted information (Zagoruyko & Komodakis, 2016; Wang et al., 2021). The reason behind the success of feature distillation is that intermediate features encode the comprehensive information during the inference process.

However, these methods share a common limitation: they directly align the intermediate features between the teacher and student models. Some studies (Cho & Hariharan, 2019; Mirzadeh et al., 2020) have demonstrated that due to differences in model capacity and feature mapping methods, there have significant discrepancies between teacher and student features, including feature space distributions and semantic levels. Therefore, directly aligning such possibly mismatched features may even lead to negative optimization of the student model, thereby degrading its performance. To alleviate the discrepancy between teacher and student, many works resort to well-designed feature processing methods (Yang et al., 2021a; Huang et al., 2023) and loss functions (Ahn et al., 2019; Tian et al., 2019a). Although these works improve the compatibility between teacher and student, the discrepancy is inherent and its optimization is still an intractable problem.

To address this issue, we hope to leverage the feature from the student itself as the supervision signals for distillation. Therefore, we further propose teacher-guided student **D**iffusion **S**elf-**KD**, dubbed as DSKD. DSKD introduces a teacher-classifier-guided diffusion model to denoise the student features, and the denoised features are regarded as the supervision target. Instead of directly aligning teacher and student features that previous works have done, DSKD performs distillation between the original and denoised student features. *The reasons why denoised student features could be seen as the distillation target are two aspects*: (1) The diffusion model is trained by teacher features to learn meaningful feature denoising capability, which is then used for student feature denoising. (2) Moreover, the denoising process of student features is also guided by the teacher

classifier. Therefore, the teacher knowledge could be implicitly transferred to the denoised student features. By the way, the denoised student features naturally do not have discrepancies with the original student features.

It is worth mentioning that a previous seminal work called DiffKD (Huang et al., 2023) also utilizes a diffusion model to denoise student features. Compared to DiffKD, our DSKD differs in three critical aspects: (1) **Diffusion model**. DiffKD uses a conventional diffusion model without extra optimization. By contrast, our DSKD adopts *teacher-classifier-guided* diffusion model for student feature sampling, emphasizing class-related information from the teacher network, which is beneficial to especially visual recognition tasks. (2) **Distillation supervision**. DiffKD still follows the traditional teacher-student alignment paradigm, leading to a potential discrepancy problem, as discussed above. Moreover, the denoised student features often have encoded teacher knowledge, therefore aligning the former with latter may *cancel out* those common information, leading to the information loss for the original student features. By contrast, our DSKD regards the denoised student features as the virtual teacher role to distill the original student features, avoiding the gap of feature distributions between teacher and student while learning the teacher's knowledge effectively. (3) **Distillation loss**. Compared to the traditional Mean Squared Error (MSE) loss, we propose locality-sensitive hashing (Datar et al., 2004) (LSH)-guided feature distillation that emphasizes more on feature direction than magnitude. We also provide theoretical proofs to justify the effectiveness of LSH.

We conduct image classification and semantic segmentation experiments to evaluate our method. Experimental results show that our DSKD achieves the best performance compared to recent state-of-the-art distillation methods on both homogeneous and heterogeneous architectures. Visualization results further demonstrate that the student learns similar feature patterns with the teacher even if we do not guide the student to mimic the teacher directly.

Our contributions are mainly divided into three aspects: (1) We introduce *teacher-classifier-guided* diffusion model to denoise the student features and augment them with class-related information. (2) We propose LSH-guided self-KD to avoid negative optimization towards teacher-student discrepancy, and force the student to focus on align feature direction. (3) DSKD achieves the best performance across classification, detection, and segmentation tasks.

## 2. Related Work

### 2.1. Knowledge Distillation

Knowledge Distillation (KD) aims to transfer meaningful knowledge from a powerful teacher to a weak student,

therefore improving the student performance. The original KD (Hinton et al., 2015) aligns the final class probability distributions between teacher and student, which is intuitive yet effective. This way ignores the intermediate feature information among the hidden layers. Therefore, many works explore feature-based distillation to provide more comprehensive guidance.

The seminal FitNet (Romero et al., 2014) proposes to align teacher and student intermediate features in a layer-by-layer manner, making the student learn the feature extraction process from the teacher. However, this method does not consider higher-level feature modeling to explore more advanced feature knowledge forms. To address this issue, many works attempt to mine richer knowledge encoded in the original feature maps, such as attention maps (Zagoruyko & Komodakis, 2016; Guo et al., 2023; Pham et al., 2024), relationships (Park et al., 2019; Yang et al., 2022b), contrastive representations (Tian et al., 2019a; Zhu et al., 2021; Yang et al., 2023), multi-scale fusion (Chen et al., 2021; Wei et al., 2024), and so on. Recently, DiffKD (Huang et al., 2023) also introduces a diffusion model to assist the distillation process. Our DSKD outperforms DiffKD by a more advanced *teacher-classifier-guided* diffusion model and a student-friendly self-distillation loss.

### 2.2. Diffusion Model

Denoising Diffusion Probabilistic Model (DDPM) (Ho et al., 2020) has become a well-known generative paradigm for high-quality image generation. It often includes a forward and reverse process. During the forward process, the diffusion model progressively adds noise through multiple steps starting from a real image and obtains pure randomness like Gaussian noise. During the reverse process, the diffusion model learns to reverse this process by predicting and removing noise step-by-step, reconstructing the original image from randomness. Song *et al.* (Song et al., 2020) proposed Denoising Diffusion Implicit Model (DDIM) that constructs non-Markovian diffusion processes to obtain deterministic generation, leading to faster generation. Dhariwal *et al.* (Dhariwal & Nichol, 2021) proposed to improve diffusion models with classifier guidance for a better trade-off between diversity and fidelity using gradients from an extra classifier. More detailed discussions could refer to Croitoru *et al.*'s survey (Croitoru et al., 2023).

Recently, some distillation works (Salimans & Ho, 2022; Sun et al., 2023; Feng et al., 2024) are proposed to accelerate the diffusion model by reducing the number of iterative steps. Their core idea is to regard the original diffusion model with complete steps as the teacher, and the counterpart with reduced steps as the student. The distillation loss guides the student to match the teacher's denoised trajectory. Therefore, the distilled student could generate high-quality

*Figure 1.* The overview of our proposed DSKD. We design a highly efficient yet effective diffusion backbone by combining the advantages of U-Net (Ronneberger et al., 2015) and diffusion transformer (DiT) (Peebles & Xie, 2023). Inspired by DiT, we apply a Multi-Layer Perceptron (MLP) to regress scale and shift parameters for batch normalization layer, and scaling parameters before residual summation, from the conditioned timestep embedding. Inspired by U-Net, we first adopt a $3 \times 3$ convolution with stride $S = 2$ for downsampling and then a $3 \times 3$ deconvolution with stride $S = 2$ for upsampling. The final noise prediction is formulated as a residual output. **We emphasize that the diffusion backbone is very light-weight, which only has 0.82M parameters and 85M FLOPs for ImageNet.**

images using only very few steps. **Beyond distilling diffusion model itself, using diffusion model to improve visual recognition distillation is also a promising direction but highly ignored.**

## 3. Methodology

We introduce teacher-classifier-guided diffusion model to denoise the student features. In this way, the teacher model's knowledge can be implicitly transferred to the student features. We then align the original student features to the denoised student features. This allows the student model to learn from the teacher without considering the negative discrepancy. The overview of DSKD is shown in Figure 1.

### 3.1. Preliminary: Diffusion Model

**Forward noising process.** Given a sample $\boldsymbol{x}_0 \sim q(\boldsymbol{x}_0)$ from the data distribution, the forward process progressively adds Gaussian noise for $T$ diffusion steps according to the noise schedule $\beta_1, \cdots, \beta_T$:

$$q(\boldsymbol{x}_t|\boldsymbol{x}_{t-1}) = \mathcal{N}(\boldsymbol{x}_t; \sqrt{1 - \beta_t}\boldsymbol{x}_{t-1}, \beta_t\boldsymbol{I}). \quad (1)$$

Finally, $\boldsymbol{x}_T \sim \mathcal{N}(\mathbf{0}, \boldsymbol{I})$ is pure Gaussian noise. A noisy sample $\boldsymbol{x}^t$ could be formulated as a one-step equation:

$$q(\boldsymbol{x}_t|\boldsymbol{x}_0) = \mathcal{N}(\boldsymbol{x}_t; \sqrt{\bar{\alpha}_t}\boldsymbol{x}_0, (1 - \bar{\alpha}_t)\boldsymbol{I}). \quad (2)$$

$\boldsymbol{x}_t$ can be regarded as a linear combination of $\boldsymbol{x}_0$ and noise variable $\boldsymbol{\epsilon}_t$:

$$\boldsymbol{x}_t = \sqrt{\bar{\alpha}_t}\boldsymbol{x}_0 + \sqrt{1 - \bar{\alpha}_t}\boldsymbol{\epsilon}_t, \quad (3)$$

where $\alpha_t := 1 - \beta_t$, $\bar{\alpha}_t := \prod_{s=0}^{t} \alpha_s$ and $\boldsymbol{\epsilon}_t \sim \mathcal{N}(\mathbf{0}, \boldsymbol{I})$.

**Optimization.** During the training phase, a noise predictor $\boldsymbol{\epsilon_\theta}$ is optimized to predict the noise in $\boldsymbol{x}_t$ by minimizing the MSE loss between $\boldsymbol{x}_t$ and the original image $\boldsymbol{x}_0$:

$$\mathcal{L}_{\text{Diff}} = \mathbb{E}_{\boldsymbol{x}_0, \boldsymbol{\epsilon}_t, t}\left[\|\boldsymbol{\epsilon}_t - \boldsymbol{\epsilon_\theta}(\boldsymbol{x}_t, t)\|^2\right]. \quad (4)$$

**Reverse process.** During the test phase, $\boldsymbol{x}_0$ is reconstructed by starting from Gaussian noise $\boldsymbol{x}_T \sim \mathcal{N}(\mathbf{0}, \boldsymbol{I})$ and iteratively denoising with the trained noise predictor $\boldsymbol{\epsilon_\theta}$:

$$p_{\boldsymbol{\theta}}(\boldsymbol{x}_{t-1}|\boldsymbol{x}_t) := \mathcal{N}(\boldsymbol{x}_{t-1}; \boldsymbol{\epsilon_\theta}(\boldsymbol{x}_t, t), \sigma_t^2\boldsymbol{I}), \quad (5)$$

where $\sigma_t$ is a transition variance in DDIM. Different $\sigma_t$ values would lead to distinct generative processes. When $\sigma_t = \sqrt{(1 - \alpha_{t-1})/(1 - \alpha_t)}\sqrt{1 - \alpha_t/\alpha_{t-1}}$ for all $t$, the forward process becomes Markov chain, and the reverse process is equivalent to DDPM.

### 3.2. Teacher-Guided Student Feature Denoising

Inspired by the work *classifier-guided diffusion model* (Dhariwal & Nichol, 2021), we designed a new feature distillation method by using the teacher model to guide the reverse denoising process of the student features. Under the teacher's guidance, the distribution of the student features shifts towards that of the teacher features, enriching the student features with meaningful semantic information from the teacher.

The denoising process of classifier-guided diffusion model (Dhariwal & Nichol, 2021) is formulated as:

$$p_{\theta,\phi}(\boldsymbol{x}_t|\boldsymbol{x}_{t+1}, y) = Zp_\theta(\boldsymbol{x}_t|\boldsymbol{x}_{t+1})p_\phi(y|\boldsymbol{x}_t). \quad (6)$$

Here, the definition of each variable follows the original paper (Dhariwal & Nichol, 2021). $Z$ is a normalizing constant, $p_\theta(\boldsymbol{x}_t|\boldsymbol{x}_{t+1})$ is an unconditional reverse noising process following DDPM (Ho et al., 2020), and $p_\phi(y|\boldsymbol{x}_t)$ is a classifier, where $\boldsymbol{x}_t$ is the noise image at the $t$-th step, and $y$ is a class label.

In the context of teacher-student distillation, we formulate $p_\phi(y|\boldsymbol{x}_t)$ as the pre-trained teacher classifier, and $\boldsymbol{x}_t$ as the denoised student features at the $t$-th step. Unlike the traditional denoising formula, we condition the sampling process on the student features $\boldsymbol{f}^{(stu)} \in \mathbb{R}^{H \times W \times D}$, where $H$, $W$, and $C$ represent height, width, and the number of channels, respectively. We start $\boldsymbol{f}^{(stu)}$ as $\boldsymbol{x}_T$ to perform teacher-guided diffusion sampling by $T$ steps, *i.e.* $t = T, \cdots, 2, 1$. $\boldsymbol{x}_t$ denotes the denoised student features at the $t$-th time step. Therefore, the sampling formula of Equ.(17) can be expressed as follows:

$$p(\boldsymbol{x}_t \mid \boldsymbol{x}_{t+1}, y; \boldsymbol{\theta}, \boldsymbol{\phi}^{(tea)}) = Z p_\theta(\boldsymbol{x}_t|\boldsymbol{x}_{t+1}) p(y|\boldsymbol{x}_t; \boldsymbol{\phi}^{(tea)}). \tag{7}$$

$p(\boldsymbol{x}_t \mid \boldsymbol{x}_{t+1}, y; \boldsymbol{\theta}, \boldsymbol{\phi}^{(tea)})$ is a conditional Markov process to denoise the student feature from $\boldsymbol{x}_{t+1}$ to $\boldsymbol{x}_t$, conditioned by the noise predictor $\boldsymbol{\theta}$ and the teacher classifier $\boldsymbol{\phi}^{(tea)}$. $p(y|\boldsymbol{x}_t; \boldsymbol{\phi}^{(tea)})$ is the conditional probability of the predicted class $y$ based on the student features $\boldsymbol{x}_t$ inferenced from the teacher classifier $\boldsymbol{\phi}^{(tea)}$. The teacher classifier often includes a global average pooling layer and a linear weight matrix to output class probability distribution. We adopt the traditional diffusion model that predicts $\boldsymbol{x}_t$ from $\boldsymbol{x}_{t+1}$ according to a Gaussian distribution:

$$p_\theta(\boldsymbol{x}_t|\boldsymbol{x}_{t+1}) = \mathcal{N}(\boldsymbol{\mu}, \boldsymbol{\Sigma}), \tag{8}$$

where $\boldsymbol{\mu} = \boldsymbol{\mu}_\theta(\boldsymbol{x}_{t+1})$, $\boldsymbol{\Sigma} = \boldsymbol{\Sigma}_\theta(\boldsymbol{x}_{t+1})$. The logarithm form of Equ.(19) is formulated as:

$$\log p_\theta(\boldsymbol{x}_t|\boldsymbol{x}_{t+1}) = -\frac{1}{2}(\boldsymbol{x}_t - \boldsymbol{\mu})^\top \boldsymbol{\Sigma}^{-1}(\boldsymbol{x}_t - \boldsymbol{\mu}) + C. \tag{9}$$

When the number of diffusion steps is limited to be infinite, we can derive $\|\boldsymbol{\Sigma}\| \to \boldsymbol{0}$. In this case, $p(y|\boldsymbol{x}_t; \boldsymbol{\phi}^{(tea)})$ has low curvature compared to $\boldsymbol{\Sigma}^{-1}$. Therefore, we can approximate $\log p(y|\boldsymbol{x}_t; \boldsymbol{\phi}^{(tea)})$ by a first-order Taylor expansion at $\boldsymbol{x}_t = \boldsymbol{\mu}$:

$$\begin{aligned} \log p(y|\boldsymbol{x}_t; \boldsymbol{\phi}^{(tea)}) &\approx \log p(y|\boldsymbol{x}_t; \boldsymbol{\phi}^{(tea)})|_{\boldsymbol{x}_t=\boldsymbol{\mu}} \\ &+ (\boldsymbol{x}_t - \boldsymbol{\mu}) \nabla_{\boldsymbol{x}_t} \log p(y|\boldsymbol{x}_t; \boldsymbol{\phi}^{(tea)})|_{\boldsymbol{x}_t=\boldsymbol{\mu}} \\ &= (\boldsymbol{x}_t - \boldsymbol{\mu})\boldsymbol{g} + C_1, \end{aligned} \tag{10}$$

where $\boldsymbol{g} = \nabla_{\boldsymbol{x}_t} \log p(y|\boldsymbol{x}_t; \boldsymbol{\phi}^{(tea)})|_{\boldsymbol{x}_t=\boldsymbol{\mu}}$, and $C_1$ can be regarded as a constant. We can further derive the logarithm form of Equ.(18) as:

$$\log(p_\theta(\boldsymbol{x}_t|\boldsymbol{x}_{t+1}) p(y|\boldsymbol{x}_t; \boldsymbol{\phi}^{(tea)})) \approx \log p(\boldsymbol{z}), \tag{11}$$

where $\boldsymbol{z} \sim \mathcal{N}(\boldsymbol{\mu} + \boldsymbol{\Sigma}\boldsymbol{g}, \boldsymbol{\Sigma})$. The detailed proof is shown in Appendix A.1. Therefore, the conditional sampling strategy

---

**Algorithm 1** Teacher-guided student self-knowledge distillation using diffusion model

**Input:** diffusion model $(\boldsymbol{\mu}_{\boldsymbol{\theta}}(\boldsymbol{x}_t), \boldsymbol{\Sigma}_{\boldsymbol{\theta}}(\boldsymbol{x}_t))$, pretrained teacher feature extractor $\Phi^{(tea)}$ and classifier $\phi^{(tea)}$, untrained student feature extractor $\Phi^{(stu)}$, dataset $\mathcal{D}$.

**Output:** the trained student model.

1: **for** each $(\boldsymbol{x}, y) \in \mathcal{D}$ **do**
2:     Extract features: $\boldsymbol{f}^{(tea)} = \Phi^{(tea)}(\boldsymbol{x})$, $\boldsymbol{f}^{(stu)} = \Phi^{(stu)}(\boldsymbol{x})$
3:     Train the diffusion model $(\boldsymbol{\mu}_{\boldsymbol{\theta}}(\boldsymbol{x}_t), \boldsymbol{\Sigma}_{\boldsymbol{\theta}}(\boldsymbol{x}_t))$ using $\boldsymbol{f}^{(tea)}$ by $\mathcal{L}_{\text{Diff}}$ (Equ.(4)).
4:     Start at $\boldsymbol{f}^{(stu)}$ as $\boldsymbol{x}_T$ to perform teacher-guided diffusion denoising sampling.
5:     **for all** $t$ from $T$ to 1 **do**
6:         $\boldsymbol{\mu}, \boldsymbol{\Sigma} \leftarrow \boldsymbol{\mu}_{\boldsymbol{\theta}}(\boldsymbol{x}_t), \boldsymbol{\Sigma}_{\boldsymbol{\theta}}(\boldsymbol{x}_t)$
7:         $\boldsymbol{x}_{t-1} \sim \mathcal{N}(\boldsymbol{\mu} + k\boldsymbol{\Sigma} \nabla_{\boldsymbol{x}_t} \log p(y|\boldsymbol{x}_t; \boldsymbol{\phi}^{(tea)}), \boldsymbol{\Sigma})$
8:     **end for**
9:     Compute the DSKD loss $\mathcal{L}_{\text{DSKD}}$ (Equ.(15)) between the original and denoised student features of $\boldsymbol{f}^{(stu)}$ and $\hat{\boldsymbol{f}}^{(stu)}$ (*i.e.* $\boldsymbol{x}_0$).
10:    Compute the basic task loss $\mathcal{L}_{\text{Task}}$ by cross-entropy.
11:    Compute the Hinton's distillation loss $\mathcal{L}_{\text{KD}}$.
12:    Update the student model by optimizing $\mathcal{L}_{\text{Train}}$.
13: **end for**

---

can be approximated to the unconditional Gaussian sampling, but differs in the shifted mean by $\boldsymbol{\Sigma}\boldsymbol{g}$. Moreover, we introduce a gradient scale $k$ as the guidance strength over the gradient of the teacher classifier. In summary, the teacher-guided diffusion sampling is formulated as:

$$\boldsymbol{x}_{t-1} \sim \mathcal{N}(\boldsymbol{\mu} + k\boldsymbol{\Sigma} \nabla_{\boldsymbol{x}_t} \log p(y|\boldsymbol{x}_t; \boldsymbol{\phi}^{(tea)}), \boldsymbol{\Sigma}). \tag{12}$$

In theory, the gradient scale $k$ can smooth the teacher class probability distribution, proportional to $p(y|\boldsymbol{x}_t; \boldsymbol{\phi}^{(tea)})^k$. When $k > 1$, the distribution becomes sharper, meaning that the teacher classifier has stronger guidance strength, resulting in higher fidelity (but less diverse) image features.

After the teacher-guided diffusion sampling process proceeds by $T$ steps, the original student features $\boldsymbol{f}^{(stu)}$ (*i.e.* $\boldsymbol{x}_T$) is converted to the denoised student features $\hat{\boldsymbol{f}}^{(stu)}$ (*i.e.* $\boldsymbol{x}_0$). According to Table 10, we set $T = 2$ and $T = 3$ on CIFAR-100 and ImageNet, respectively.

### 3.3. LSH-Guided Student Self-Feature Distillation

Our method uses classifier-guided diffusion model as the generator for sampling student features, making the distribution of denoised student feature $\hat{\boldsymbol{f}}^{(stu)}$ more similar to that of the teacher feature $\boldsymbol{f}^{(tea)}$. This lets the denoised student feature $\hat{\boldsymbol{f}}^{(stu)}$ learn similar semantic information and representation capability from the teacher feature $\boldsymbol{f}^{(tea)}$

*Table 1.* Top-1 accuracy of different distillation methods under homogeneous and heterogeneous architecture styles on CIFAR-100 dataset. We follow the training strategy A1 in Table 9.

| Method | Homogeneous architecture style | | | Heterogeneous architecture style | | |
|---|---|---|---|---|---|---|
| | WRN-40-2 WRN-40-1 | ResNet-56 ResNet-20 | ResNet-32x4 ResNet-8x4 | ResNet-56 WRN-40-1 | WRN-40-2 ResNet-20 | ResNet-32x4 ResNet-20 |
| Teacher | 75.61 | 72.34 | 79.42 | 72.34 | 75.61 | 79.42 |
| Student | 71.98 | 69.06 | 72.50 | 71.98 | 69.06 | 69.06 |
| KD (Hinton et al., 2015) | 73.54±0.20 | 70.66±0.24 | 73.33±0.20 | 73.39±0.17 | 71.15±0.22 | 70.21±0.09 |
| FitNet (Romero et al., 2014) | 72.24±0.24 | 69.21±0.36 | 73.50±0.28 | 72.22±0.28 | 69.34±0.20 | 69.81±0.07 |
| VID (Ahn et al., 2019) | 73.30±0.13 | 70.38±0.14 | 73.09±0.21 | 73.37±0.16 | 70.41±0.18 | 70.35±0.11 |
| RKD (Park et al., 2019) | 72.22±0.20 | 69.61±0.06 | 71.90±0.11 | 72.16±0.23 | 69.59±0.18 | 69.76±0.18 |
| PKT (Passalis et al., 2020) | 73.45±0.19 | 70.34±0.20 | 73.64±0.18 | 73.23±0.19 | 69.92±0.13 | 70.28±0.20 |
| CRD (Tian et al., 2019a) | 74.14±0.22 | 71.16±0.17 | 75.51±0.18 | 74.06±0.20 | 71.03±0.19 | 71.19±0.14 |
| CTKD (Li et al., 2023) | 73.93±0.17 | 71.19±0.28 | 76.44±0.32 | 73.68±0.15 | 71.24±0.26 | 71.18±0.33 |
| DiffKD (Huang et al., 2023) | 74.09±0.09 | 71.92±0.14 | 76.72±0.15 | 73.99±0.13 | 71.60±0.27 | 71.39±0.21 |
| CAT-KD (Guo et al., 2023) | 74.26±0.07 | 71.64±0.15 | 76.88±0.09 | 73.68±0.21 | 71.58±0.13 | 71.21±0.06 |
| LS (Sun et al., 2024) | 74.33±0.08 | 71.43±0.16 | 76.59±0.11 | 73.71±0.24 | 71.47±0.28 | 71.27±0.33 |
| SD (Wei et al., 2024) | 73.86±0.13 | 71.35±0.27 | 76.53±0.18 | 73.83±0.34 | 71.42±0.14 | 71.30±0.20 |
| IKD (Wang et al., 2025) | 74.16±0.11 | 71.83±0.23 | 76.58±0.17 | 73.47±0.35 | 71.28±0.29 | 70.89±0.14 |
| DSKD (Ours) | **74.45±0.16** | **72.26±0.12** | **77.08±0.10** | **74.70±0.07** | **72.13±0.19** | **71.63±0.14** |

implicitly. Moreover, since the teacher model often has powerful discriminative capability, the denoised student feature $\hat{\boldsymbol{f}}^{(stu)}$ could preserve class-related information during the teacher-guided diffusion sampling process compared to the conventional teacher-agnostic DiffKD (Huang et al., 2023).

In the context of the distillation process, student features are considered to be the "noisy version" of teacher model features, as demonstrated by DiffKD (Huang et al., 2023). We also introduce a *noise adapter* to initialize the noise level of student features, whose details are shown in Appendix A.2. After the teacher-guided student feature diffusion sampling process, the original student feature $\boldsymbol{f}^{(stu)}$ would be transformed to the teacher-guided denoised student feature $\hat{\boldsymbol{f}}^{(stu)}$.

Unlike most previous KD methods including DiffKD that use the teacher to distill the student, our DSKD method utilizes the denoised student feature $\hat{\boldsymbol{f}}^{(stu)}$ to distill the original student feature $\boldsymbol{f}^{(stu)}$ via comprehensive local and global distillation. The local distillation follows a traditional paradigm by aligning the full feature-maps via a simple MSE loss:

$$\mathcal{L}_{\text{Local}} = \left\| \boldsymbol{f}^{(stu)} - \hat{\boldsymbol{f}}^{(stu)} \right\|_2^2, \tag{13}$$

which makes the student learn spatially dense semantic information. On the other hand, we further apply global average pooling to $\boldsymbol{f}^{(stu)}$ and $\hat{\boldsymbol{f}}^{(stu)}$, and output $\boldsymbol{v}^{(stu)} \in \mathbb{R}^D$ and

$\hat{\boldsymbol{v}}^{(stu)} \in \mathbb{R}^D$ as global feature embeddings, respectively. Unlike local features that may contain noisy information, the global features encode robust object-centric representations. And then we construct a novel locality-sensitive hashing (Datar et al., 2004) (LSH) Guided global distillation method for aligning $\boldsymbol{v}^{(stu)}$ and $\hat{\boldsymbol{v}}^{(stu)}$. In this approach, we adopt $M$ hash functions, and construct $M$ binary codes for denoised student feature $\hat{\boldsymbol{v}}^{(stu)}$. The hash code of original student feature $\boldsymbol{v}^{(stu)}$ is required to be identical to that of denoised student feature $\hat{\boldsymbol{v}}^{(stu)}$. Therefore, we formulate this problem as minimizing a binary cross-entropy classification loss:

$$\mathcal{L}_{\text{Global}} = -\frac{1}{M} \sum_{m=1}^{M} \left[ \delta_m \log \rho_m + (1 - \delta_m) \log(1 - \rho_m) \right], \tag{14}$$

where $\boldsymbol{\delta} = \text{sign}(\boldsymbol{W}^\top \hat{\boldsymbol{v}}^{(stu)} + \boldsymbol{b}) \in \mathbb{R}^M$, $\boldsymbol{\rho} = \text{Sigmoid}(\boldsymbol{W}^\top \boldsymbol{v}^{(stu)} + \boldsymbol{b}) \in \mathbb{R}^M$, $\boldsymbol{W} \in \mathbb{R}^{D \times M}$ represents projection matrix whose values are sampled from a Gaussian distribution, and $\boldsymbol{b} \in \mathbb{R}^M$ denotes the bias. We set $M = 256$ according to Appendix B.3. Inspired by FNKD (Xu et al., 2020), feature direction is more important than its magnitude. LSH can also achieve this goal that relaxes the constraints of feature distillation on magnitude, but prioritize feature direction alignment. Detailed justification and theoretical analyses are shown in Appendix A.3 and A.4.

The overall loss of DSKD is formulated as the summation

of $\mathcal{L}_{\text{Local}}$ and $\mathcal{L}_{\text{Global}}$:

$$\mathcal{L}_{\text{DSKD}} = \mathcal{L}_{\text{Local}} + \gamma \mathcal{L}_{\text{Global}}, \qquad (15)$$

where $\gamma$ is a loss weight and we found $\gamma = 1$ works well. In this way, DSKD can guide the student to learn meaningful knowledge from the more powerful teacher indirectly through a diffusion model. Notice that the conventional KD methods often have the representational gap problem between teacher and student, and they often resort to complex training schemes, loss functions, and feature alignment techniques to alleviate this issue. By contrast, our DSKD eliminates the inherent difference of teacher-student feature distributions and guides the student to learn compatible information with the teacher.

### 3.4. Overall Training Process

The overview of the proposed DSKD is shown in Algorithm 1. First, we extract teacher and student features $\boldsymbol{f}^{(tea)}$ and $\boldsymbol{f}^{(stu)}$. We regard $\boldsymbol{f}^{(tea)}$ as $\boldsymbol{x}_0$ in the forward noise process $q(\boldsymbol{x}_t|\boldsymbol{x}_0)$ (Equ.(2)) and then train the teacher-guided diffusion model by $\mathcal{L}_{\text{Diff}}$ (Equ.(4)). We start at $\boldsymbol{f}^{(stu)}$ as $\boldsymbol{x}_T$ to perform teacher-guided diffusion denoising sampling by Equ.(23) with $T$ time steps. After the denoising process, the denoised student feature $\hat{\boldsymbol{f}}^{(stu)}$ is produced. We compute the DSKD loss (Equ.(13)) between the original and denoised student features of $\boldsymbol{f}^{(stu)}$ and $\hat{\boldsymbol{f}}^{(stu)}$. Moreover, the task loss $\mathcal{L}_{\text{Task}}$ is the conventional cross-entropy loss between the student class probability and ground-truth label. The $\mathcal{L}_{\text{KD}}$ loss is KL-divergence between teacher and student class probability distributions following by Hinton *et al.* (Hinton et al., 2015). In summary, the overall loss is formulated as:

$$\mathcal{L}_{\text{Train}} = \mathcal{L}_{\text{Task}} + \alpha \mathcal{L}_{\text{DSKD}} + \mathcal{L}_{\text{Diff}} + \mathcal{L}_{\text{KD}}, \qquad (16)$$

where $\alpha$ is the DSKD loss weight, and we set $\alpha = 1$ according to Figure 3. We do not introduce extra weights for the basic $\mathcal{L}_{\text{Task}}$, $\mathcal{L}_{\text{Diff}}$, and $\mathcal{L}_{\text{KD}}$ following previous works (Yang et al., 2021a; 2022b; Huang et al., 2023).

## 4. Experiments

### 4.1. Experimental Setup

(1) **Datasets.** We adopt CIFAR-100 (Krizhevsky & Hinton, 2009) and ImageNet (Deng et al., 2009) datasets for image classification, COCO (Lin et al., 2014) for object detection, ADE20K (Zhou et al., 2017) for semantic segmentation. Please see details in B.1. (2) **Network architecture.** We use famous network families for evaluation, including ResNet (He et al., 2016), WRN (Zagoruyko S, 2016), MobileNetV1 (Howard et al., 2017), MobileNetV3 (Howard et al., 2019), Swin Transformer (Liu et al., 2021), and DeepLabV3 (Chen et al., 2018). (3) **Compared methods** We compare our method with recent representative

teacher-student distillation methods, including KD (Hinton et al., 2015), Review (Chen et al., 2021), DKD (Zhao et al., 2022), Manifold (Hao et al., 2022), CTKD (Li et al., 2023), DiffKD (Huang et al., 2023), CAT-KD (Guo et al., 2023), LS (Sun et al., 2024), SD (Wei et al., 2024), IKD (Wang et al., 2025), and RLD (Sun et al., 2025). Moreover, we also include self-distillation methods, including BYOT (Zhang et al., 2019), Tf-KD (Yuan et al., 2020), CS-KD (Yun et al., 2020), PS-KD (Kim et al., 2021), FRSKD (Ji et al., 2021), DLB (Shen et al., 2022), MixSKD (Yang et al., 2022a), FASD (Xu et al., 2024).

### 4.2. Experimental Results on Image Classification

Table 1 shows the compared results on CIFAR-100. Compared with state-of-the-art IKD, our DSKD achieves 0.40% and 0.94% average improvements on homogeneous and heterogeneous architecture pairs, respectively. As shown in Table 2, DSKD also performs the best on the large-scale ImageNet dataset. DSKD outperforms IKD on ResNet-18 and MobileNetV1 by 0.75% and 0.46% top-1 accuracy gains, respectively. Table 4 exhibits the ImageNet results on Swin Transformer. DSKD obtains the highest 82.8% accuracy on Swin-Tiny, surpassing DiffKD and IKD by 0.4% and 0.8%, respectively. The results verify that our DSKD could well adapt to vision Transformer architecture. Overall results demonstrate that teacher-guided diffusion denoising is an effective way to student feature self-distillation.

### 4.3. Results on Segmentation and Detection

As shown in Table 5, we apply our DSKD to object detection and compare some advanced detection distillation methods on COCO dataset (Lin et al., 2014). We follow the same training settings as DiffKD (Huang et al., 2023). DSKD achieves the best performance consistently across various detectors. It surpasses the second-best DiffKD by 0.8%, 0.8%, and 0.4% AP on two-stage detector Faster-RCNN (Ren et al., 2015), one-stage detector RetinaNet (Lin et al., 2017), and anchor-free detector FCOS (Tian et al., 2019b), respectively.

Table 6 shows the extended results for semantic segmentation on ADE20K, where the training settings are followed by CIRKD (Yang et al., 2022b). DSKD also achieves the best performance and exceeds state-of-the-art DiffKD and LAD by 1.73% and 1.97% mIoU improvements, respectively. The results demonstrate that DSKD can be well transferred to the downstream semantic segmentation task.

### 4.4. Comparison with Self-Distillation methods

Although our DSKD is a teacher-student-based distillation method, we also compare those teacher-free self-distillation methods in Table 3. DSKD achieves the best performance

*Table 2.* Accuracy results on ImageNet following the training strategy B1 in Table 9.

| Teacher | Student | Metric | Tea. | Stu. | KD | Review | DKD | CTKD | DiffKD | CAT-KD | LS | SD | IKD | RLD | DSKD |
|---------|---------|--------|------|------|-----|--------|------|------|--------|--------|------|------|------|------|------|
| ResNet-34 | ResNet-18 | Top-1 | 73.31 | 69.76 | 70.66 | 71.61 | 71.70 | 71.32 | 72.22 | 71.26 | 71.42 | 71.44 | 71.82 | 71.91 | **72.57** |
|  |  | Top-5 | 91.42 | 89.08 | 89.88 | 90.51 | 90.41 | 90.27 | 90.64 | 90.45 | 90.29 | 90.05 | 90.57 | 90.59 | **90.82** |
| ResNet-50 | MobileNetV1 | Top-1 | 76.16 | 70.13 | 70.68 | 72.56 | 72.05 | 72.87 | 73.27 | 72.24 | 72.18 | 72.24 | 73.27 | 72.75 | **73.73** |
|  |  | Top-5 | 92.86 | 89.49 | 90.30 | 91.00 | 91.05 | 91.29 | 91.14 | 91.13 | 90.80 | 90.71 | 91.34 | 91.18 | **91.52** |

*Table 3.* Top-1 accuracy results of ResNet-18 on ImageNet compared with recent self-distillation methods.

| Method | Stu. | BYOT | Tf-KD | CS-KD | PS-KD | FRSKD | DLB | MixSKD | FASD | DSKD |
|--------|------|------|-------|-------|-------|-------|-----|--------|------|------|
| Acc | 69.76 | 71.34 | 70.44 | 70.65 | 70.96 | 71.36 | 70.93 | 71.52 | 71.43 | **72.57** |

*Table 4.* Top-1 accuracy results of Swin Transformer on ImageNet following the training strategy B2 in Table 9.

| Teacher | Student | Tea. | Stu. | KD | Manifold | DiffKD | IKD | DSKD |
|---------|---------|------|------|-----|----------|--------|-----|------|
| Swin-Base | Swin-Tiny | 83.5 | 81.3 | 81.8 | 82.3 | 82.4 | 82.0 | **82.8** |

*Table 5.* Object detection distillation on COCO validation set.

| Method | AP | $AP_{50}$ | $AP_{75}$ | $AP_S$ | $AP_M$ | $AP_L$ |
|--------|-----|-----------|-----------|--------|--------|--------|
| *Two-stage detectors* | | | | | | |
| T: Faster RCNN-R101 | 39.8 | 60.1 | 43.3 | 22.5 | 43.6 | 52.8 |
| S: Faster RCNN-R50 | 38.4 | 59.0 | 42.0 | 21.5 | 42.1 | 50.3 |
| FitNet (Romero et al., 2014) | 38.9 | 59.5 | 42.4 | 21.9 | 42.2 | 51.6 |
| FRS (Du et al., 2021) | 39.5 | 60.1 | 43.3 | 22.3 | 43.6 | 51.7 |
| FGD (Yang et al., 2021b) | 40.4 | - | - | 22.8 | 44.5 | 53.5 |
| DiffKD (Huang et al., 2023) | 40.7 | 61.0 | 44.3 | 22.6 | 44.6 | 53.7 |
| DSKD (Ours) | **41.5** | **61.4** | **44.9** | **23.3** | **45.2** | **54.4** |
| *One-stage detectors* | | | | | | |
| T: RetinaNet-R101 | 38.9 | 58.0 | 41.5 | 21.0 | 42.8 | 52.4 |
| S: RetinaNet-R50 | 37.4 | 56.7 | 39.6 | 20.0 | 40.7 | 49.7 |
| FitNet (Romero et al., 2014) | 37.4 | 57.1 | 40.0 | 20.8 | 40.8 | 50.9 |
| FRS (Du et al., 2021) | 39.3 | 58.8 | 42.0 | 21.5 | 43.3 | 52.6 |
| FGD (Yang et al., 2021b) | 39.6 | - | - | 22.9 | 43.7 | 53.6 |
| DiffKD (Huang et al., 2023) | 39.8 | 58.7 | 42.5 | 21.5 | 43.6 | 53.2 |
| DSKD (Ours) | **40.6** | **59.4** | **42.8** | **23.1** | **44.2** | **53.9** |
| *Anchor-free detectors* | | | | | | |
| T: FCOS-R101 | 40.8 | 60.0 | 44.0 | 24.2 | 44.3 | 52.4 |
| S: FCOS-R50 | 38.5 | 57.7 | 41.0 | 21.9 | 42.8 | 48.6 |
| FRS (Du et al., 2021) | 40.9 | 60.3 | 43.6 | 25.7 | 45.2 | 51.2 |
| FGD (Yang et al., 2021b) | 42.1 | - | - | **27.0** | 46.0 | 54.6 |
| DiffKD (Huang et al., 2023) | 42.5 | 61.1 | 45.6 | 25.2 | 46.8 | 55.1 |
| DSKD (Ours) | **42.9** | **61.7** | **45.7** | 26.8 | **47.3** | **55.8** |

*Table 6.* Performance comparison with semantic segmentation distillation methods on ADE20K.

| Method | mIoU (%) |
|--------|----------|
| T: DeepLabV3-ResNet-101 | 43.83 |
| S: DeepLabV3-MobileNetV3-Large | 32.83 |
| +SKD (Liu et al., 2019) | 33.78 |
| +IFVD (Wang et al., 2020) | 34.21 |
| +CWD (Shu et al., 2021) | 34.18 |
| +DSD (Feng et al., 2021) | 33.64 |
| +CIRKD (Yang et al., 2022b) | 35.15 |
| +APD (Tian et al., 2022) | 34.53 |
| +Af-DCD (Fan et al., 2023) | 35.12 |
| +DiffKD (Huang et al., 2023) | 34.85 |
| +LAD (Liu et al., 2024) | 34.61 |
| +DSKD (Ours) | **36.58** |

*Table 7.* Ablation study of distillation loss terms over ResNet-18 supervised by ResNet-34 on ImageNet.

| Loss | Different loss combinations | | | | | |
|------|------|------|------|------|------|------|
| $\mathcal{L}_{Task}$ | ✓ | ✓ | ✓ | ✓ | ✓ | ✓ |
| $\mathcal{L}_{Local}$ | - | ✓ | - | ✓ | - | ✓ |
| $\mathcal{L}_{Global}$ | - | - | ✓ | ✓ | - | ✓ |
| $\mathcal{L}_{Diff}$ | - | ✓ | ✓ | ✓ | - | ✓ |
| $\mathcal{L}_{KD}$ | - | - | - | - | ✓ | ✓ |
| Accuracy | 69.76 | 71.33 | 71.85 | 72.17 | 70.66 | **72.57** |

and outperforms SOTA MixSKD by 1.05%. A weakness of teacher-free self-distillation is the limited performance upper bound due to the lack of a teacher. By contrast, our DSKD constructs target features denoised by a teacher-guided diffusion model, making the student learn from the teacher.

### 4.5. Ablation Study and Analysis

By default, the experimental teacher-student pair is WRN-40-2&WRN-40-1 for CIFAR-100 and ResNet-34&ResNet-18 for ImageNet.

**Analysis of guidance strengths.** The guidance strength refers to the hyper-parameter $k$ to reweight the gradient of the teacher classifier $\nabla_{\boldsymbol{x}_t} \log p(y|\boldsymbol{x}_t; \boldsymbol{\phi}^{(tea)})$ during the diffusion denoising process. When $k > 1$, the distribution becomes sharper, resulting in higher fidelity (but less diverse) image features. As shown in Figure 2, we investigated the effects of different guidance strengths on CIFAR-100 and ImageNet datasets. We found that $k = 1$ and $k = 2$ achieve the best performance on CIFAR-100 and ImageNet, respectively. Lower $k$ may weaken the strength of teacher guidance, while higher $k$ would decrease the diversity of image features. Moreover, we found that ImageNet needs a larger $k$ than CIFAR-100, because ImageNet is a more diverse and higher resolution image dataset than CIFAR-100.

**Effect of DSKD Loss.** To evaluate the impact of the proposed DSKD loss term ($\mathcal{L}_{Local}$ and $\mathcal{L}_{Global}$), we conducted

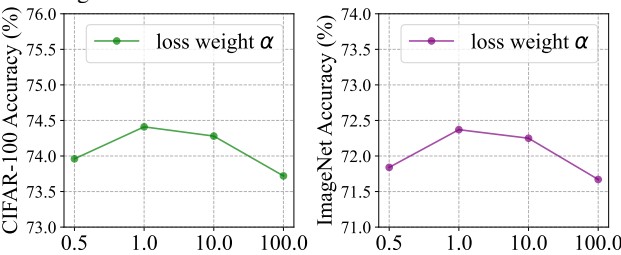

*Figure 2.* Analysis of various guidance strengths on CIFAR-100 and ImageNet datasets.

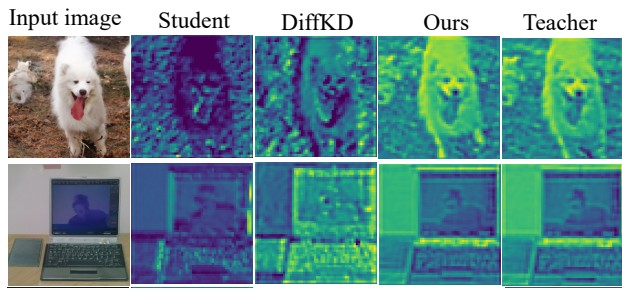

*Figure 4.* Visualizations of the original student features, DiffKD's denoised student features, our denoised student features, and teacher features on ImageNet.

*Table 8.* Training costs per epoch of ResNet-18 on ImageNet

| Method | Params | FLOPs | Time | GPU memory | Acc. |
|---|---|---|---|---|---|
| KD | 11.7M | 1.81G | 15min | 2176MB | 70.66 |
| ReviewKD | 30.9M | 3.71G | 36min | 6823MB | 71.61 |
| DiffKD | 13.6M | 2.61G | 31min | 3275MB | 72.22 |
| DSKD | 12.5M | 2.07G | 21min | 2472MB | **72.57** |

*Figure 3.* Analysis of the weight $\alpha$ of DSKD loss $\mathcal{L}_{\text{DSKD}}$ on CIFAR-100 and ImageNet datasets.

an ablation study on ImageNet in Table 7. When only the KD loss ($\mathcal{L}_{\text{KD}}$) is applied, the student model achieves an accuracy of 70.66%, which serves as the distillation baseline for comparison. By replacing $\mathcal{L}_{\text{KD}}$ with DSKD-based $\mathcal{L}_{\text{Local}}$ and $\mathcal{L}_{\text{Global}}$, the accuracy increases to 72.17%, demonstrating that the DSKD loss is more effective than the traditional KD loss. Moreover, $\mathcal{L}_{\text{Global}}$ outperforms $\mathcal{L}_{\text{Local}}$ by 0.52%, indicating that LSH-guided global distillation is more important. Finally, when all losses of $\mathcal{L}_{\text{KD}}$, $\mathcal{L}_{\text{DSKD}}$, and $\mathcal{L}_{\text{Diff}}$ are combined, the accuracy is further boosted to 72.57%. This demonstrates the effectiveness of using denoised student features as the distillation supervision.

**Analysis of the weight $\alpha$ of DSKD loss.** As shown in Figure 3, we examine the sensitivity of the loss weight $\alpha$ within [0.5, 100]. We found that $\alpha \in [1, 10]$ achieves good performance, and $\alpha = 1$ performs the best on both CIFAR-100 and ImageNet.

**Visualization of denoised features.** As shown in Figure 4, we visualize the original student features, denoised student features by DiffKD and our DSKD, teacher features during the distillation process. Compared to the original and DiffKD student features, we found that our denoised student features have more similar semantics and salient distributions with the teacher features. The results reveal that our method can remove noise information over the original features effectively. Our teacher-classifier-guided diffusion process emphasizes more class-related semantics than DiffKD, therefore leading to better feature quality.

**Efficiency analysis.** Compared to the traditional KD method, our DSKD needs an extra classifier-guided diffusion model to denoise the student features. Since the diffusion model runs 3 steps on ImageNet, the total computation of the diffusion process is 255M FLOPs. Taking the student ResNet-18 (11.7M parameters and 1.81G FLOPs) on ImageNet as an example, the extra parameters and FLOPs only accounts 7.0% and 14.1%, respectively. Moreover, the additional computation FLOPs of our DSKD are on par with or lower than popular feature-based distillation methods, such as ReviewKD (Chen et al., 2021) and DiffKD (Huang et al., 2023), which requires 1900M and 800M. We further verify the practical training costs of DSKD in Table 8. Our DSKD does not bring much training costs but leads to remarkable gains over the traditional KD.

*In Appendix, we supplement more experiments to verify the effectiveness of DSKD under various denoising timesteps, few-shot and noisy label scenarios. We also show analysis of probability distribution and attention visualization.*

## 5. Conclusion

This paper proposes DSKD, a novel KD method by introducing teacher-classifier-guided diffusion model and performing student feature self-distillation to improve visual recognition tasks. Compared to previous methods, our DSKD avoids the teacher-student discrepancy problem and augments the student features with class-related information by the teacher classifier. Experimental results on image classification and semantic segmentation show that DSKD achieves the best performance. We hope this paper will inspire future research to explore advanced KD methods by introducing diffusion models.

## Impact Statement

This paper presents work whose goal is to advance the field of machine learning. There are many potential societal consequences of our work, none of which we feel must be specifically highlighted here.

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

# A. Methodology

We introduce teacher-classifier-guided diffusion model to denoise the student features. In this way, the teacher model's knowledge can be implicitly transferred to the student features. We then align the original student features to the denoised student features. This allows the student model to learn from the teacher without considering the negative discrepancy.

## A.1. Teacher-Guided Student Feature Denoising

Inspired by the work *classifier-guided diffusion model* (Dhariwal & Nichol, 2021), we designed a new feature distillation method by using the teacher model to guide the reverse denoising process of the student features. Under the teacher's guidance, the distribution of the student features shifts towards that of the teacher features, enriching the student features with meaningful semantic information from the teacher.

The denoising process of classifier-guided diffusion model (Dhariwal & Nichol, 2021) is formulated as:

$$p_{\theta,\phi}(\boldsymbol{x}_t|\boldsymbol{x}_{t+1}, y) = Z p_\theta(\boldsymbol{x}_t|\boldsymbol{x}_{t+1}) p_\phi(y|\boldsymbol{x}_t). \tag{17}$$

Here, the definition of each variable follows the original paper (Dhariwal & Nichol, 2021). $Z$ is a normalizing constant, $p_\theta(\boldsymbol{x}_t|\boldsymbol{x}_{t+1})$ is an unconditional reverse noising process following DDPM (Ho et al., 2020), and $p_\phi(y|\boldsymbol{x}_t)$ is a classifier, where $\boldsymbol{x}_t$ is the noise image at the $t$-th step, and $y$ is a class label.

In the context of teacher-student distillation, we formulate $p_\phi(y|\boldsymbol{x}_t)$ as the pre-trained teacher classifier, and $\boldsymbol{x}_t$ as the denoised student features at the $t$-th step. Unlike the traditional denoising formula, we condition the sampling process on the student features $\boldsymbol{f}^{(stu)} \in \mathbb{R}^{H \times W \times D}$, where $H$, $W$, and $C$ represent height, width, and the number of channels, respectively. We start $\boldsymbol{f}^{(stu)}$ as $\boldsymbol{x}_T$ to perform teacher-guided diffusion sampling by $T$ steps, *i.e.* $t = T, \cdots, 2, 1$. $\boldsymbol{x}_t$ denotes the denoised student features at the $t$-th time step. Therefore, the sampling formula of Equ.(17) can be expressed as follows:

$$p(\boldsymbol{x}_t \mid \boldsymbol{x}_{t+1}, y; \boldsymbol{\theta}, \boldsymbol{\phi}^{(tea)}) = Z p_{\boldsymbol{\theta}}(\boldsymbol{x}_t|\boldsymbol{x}_{t+1}) p(y|\boldsymbol{x}_t; \boldsymbol{\phi}^{(tea)}). \tag{18}$$

$p(\boldsymbol{x}_t \mid \boldsymbol{x}_{t+1}, y; \boldsymbol{\theta}, \boldsymbol{\phi}^{(tea)})$ is a conditional Markov process to denoised the student feature from $\boldsymbol{x}_{t+1}$ to $\boldsymbol{x}_t$, conditioned by the noise predictor $\boldsymbol{\theta}$ and the teacher classifier $\boldsymbol{\phi}^{(tea)}$. $p(y|\boldsymbol{x}_t; \boldsymbol{\phi}^{(tea)})$ is the conditional probability of the predicted class $y$ based on the student features $\boldsymbol{x}_t$ inferenced from the teacher classifier $\boldsymbol{\phi}^{(tea)}$. The teacher classifier often includes a global average pooling layer and a linear weight matrix to output class probability distribution. We adopt the traditional diffusion model that predicts $\boldsymbol{x}_t$ from $\boldsymbol{x}_{t+1}$ according to a Gaussian distribution:

$$p_{\boldsymbol{\theta}}(\boldsymbol{x}_t|\boldsymbol{x}_{t+1}) = \mathcal{N}(\boldsymbol{\mu}, \boldsymbol{\Sigma}), \tag{19}$$

where $\boldsymbol{\mu} = \boldsymbol{\mu}_{\boldsymbol{\theta}}(\boldsymbol{x}_{t+1})$, $\boldsymbol{\Sigma} = \boldsymbol{\Sigma}_{\boldsymbol{\theta}}(\boldsymbol{x}_{t+1})$. The logarithm form of Equ.(19) is formulated as:

$$\log p_{\boldsymbol{\theta}}(\boldsymbol{x}_t|\boldsymbol{x}_{t+1}) = -\frac{1}{2}(\boldsymbol{x}_t - \boldsymbol{\mu})^\top \boldsymbol{\Sigma}^{-1}(\boldsymbol{x}_t - \boldsymbol{\mu}) + C. \tag{20}$$

When the number of diffusion steps is limited to be infinite, we can derive $\|\boldsymbol{\Sigma}\| \to \boldsymbol{0}$. In this case, $p(y|\boldsymbol{x}_t; \boldsymbol{\phi}^{(tea)})$ has low curvature compared to $\boldsymbol{\Sigma}^{-1}$. Therefore, we can approximate $\log p(y|\boldsymbol{x}_t; \boldsymbol{\phi}^{(tea)})$ by a first-order Taylor expansion at $\boldsymbol{x}_t = \boldsymbol{\mu}$:

$$\begin{aligned}
\log p(y|\boldsymbol{x}_t; \boldsymbol{\phi}^{(tea)}) &\approx \log p(y|\boldsymbol{x}_t; \boldsymbol{\phi}^{(tea)})|_{\boldsymbol{x}_t=\boldsymbol{\mu}} \\
&+ (\boldsymbol{x}_t - \boldsymbol{\mu}) \nabla_{\boldsymbol{x}_t} \log p(y|\boldsymbol{x}_t; \boldsymbol{\phi}^{(tea)})|_{\boldsymbol{x}_t=\boldsymbol{\mu}} \\
&= (\boldsymbol{x}_t - \boldsymbol{\mu})\boldsymbol{g} + C_1,
\end{aligned} \tag{21}$$

where $\boldsymbol{g} = \nabla_{\boldsymbol{x}_t} \log p(y|\boldsymbol{x}_t; \boldsymbol{\phi}^{(tea)})|_{\boldsymbol{x}_t=\boldsymbol{\mu}}$, and $C_1$ can be regarded as a constant. We can further derive the logarithm form of Equ.(18) as:

$$\begin{aligned}
\log(&p_{\boldsymbol{\theta}}(\boldsymbol{x}_t|\boldsymbol{x}_{t+1}) p(y|\boldsymbol{x}_t; \boldsymbol{\phi}^{(tea)})) \\
&\approx -\frac{1}{2}(\boldsymbol{x}_t - \boldsymbol{\mu})^\top \boldsymbol{\Sigma}^{-1}(\boldsymbol{x}_t - \boldsymbol{\mu}) + (\boldsymbol{x}_t - \boldsymbol{\mu})\boldsymbol{g} + C_2 \\
&= -\frac{1}{2}(\boldsymbol{x}_t - \boldsymbol{\mu} - \boldsymbol{\Sigma}\boldsymbol{g})^\top \boldsymbol{\Sigma}^{-1}(\boldsymbol{x}_t - \boldsymbol{\mu} - \boldsymbol{\Sigma}\boldsymbol{g}) + \frac{1}{2}\boldsymbol{g}^\top \boldsymbol{\Sigma}\boldsymbol{g} + C_2 \\
&= -\frac{1}{2}(\boldsymbol{x}_t - \boldsymbol{\mu} - \boldsymbol{\Sigma}\boldsymbol{g})^\top \boldsymbol{\Sigma}^{-1}(\boldsymbol{x}_t - \boldsymbol{\mu} - \boldsymbol{\Sigma}\boldsymbol{g}) + C_3 \\
&= \log p(\boldsymbol{z}) + C_4,
\end{aligned} \tag{22}$$

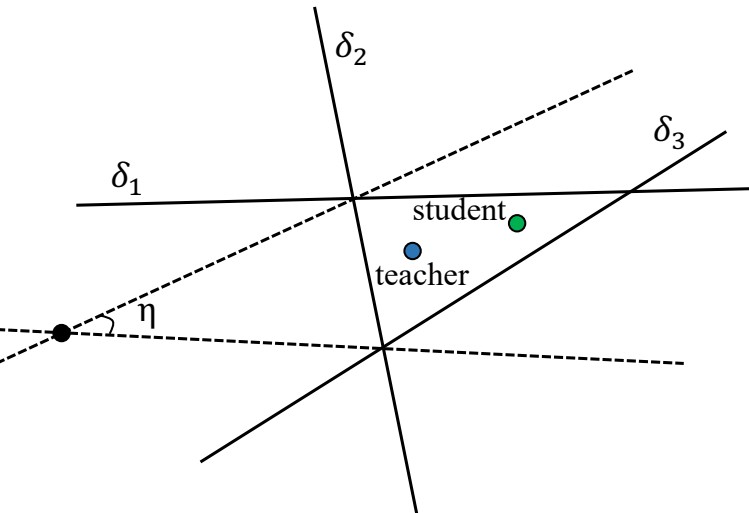

*Figure 5.* Visualization of the LSH loss mechanism. Here, $\boldsymbol{v}^{(stu)}$ and $\hat{\boldsymbol{v}}^{(stu)}$ denote the student and teacher feature vectors. The hash constraints $\delta$ define a specific polyhedral region, within which $\eta$ represents the maximal angular deviation. Note that the magnitudes of these features are allowed to vary flexibly.

where $\boldsymbol{z} \sim \mathcal{N}(\boldsymbol{\mu} + \boldsymbol{\Sigma}\boldsymbol{g}, \boldsymbol{\Sigma})$. The detailed proof is shown in Appendix A.1. Therefore, the conditional sampling strategy can be approximated to the unconditional Gaussian sampling, but differs in the shifted mean by $\boldsymbol{\Sigma}\boldsymbol{g}$. Moreover, we introduce a gradient scale $k$ as the guidance strength over the gradient of the teacher classifier. In summary, the teacher-guided diffusion sampling is formulated as:

$$\boldsymbol{x}_{t-1} \sim \mathcal{N}(\boldsymbol{\mu} + k\boldsymbol{\Sigma} \bigtriangledown_{\boldsymbol{x}_t} \log p(y|\boldsymbol{x}_t; \boldsymbol{\phi}^{(tea)}), \boldsymbol{\Sigma}). \tag{23}$$

In theory, the gradient scale $k$ can smooth the teacher class probability distribution, proportional to $p(y|\boldsymbol{x}_t; \boldsymbol{\phi}^{(tea)})^k$. When $k > 1$, the distribution becomes sharper, meaning that the teacher classifier has stronger guidance strength, resulting in higher fidelity (but less diverse) image features.

After the teacher-guided diffusion sampling process proceeds by $T$ steps, the original student features $\boldsymbol{f}^{(stu)}$ (*i.e.* $\boldsymbol{x}_T$) is converted to the denoised student features $\tilde{\boldsymbol{f}}^{(stu)}$ (*i.e.* $\boldsymbol{x}_0$).

### A.2. Details of Noise Adapter

Consistent with our previous discussion, we model the student feature as a noisy approximation of the teacher. However, the magnitude of this noise—reflecting the discrepancy between the two—is latent and varies across different training instances. This variability precludes the direct identification of the optimal initial diffusion timestep. To overcome this hurdle, we propose an adaptive noise matching module designed to calibrate the student feature's noise intensity to a pre-determined level.

We employ a lightweight convolutional block to regress a mixing coefficient $\kappa$, which blends the student output with Gaussian noise. This operation is designed to calibrate the student's noise magnitude to match the specific intensity required at the initial timestep $T$. Consequently, the starting feature for the denoising trajectory is formulated as:

$$\hat{\boldsymbol{f}}_T^{(stu)} = \kappa \boldsymbol{f}^{(stu)} + (1 - \kappa)\boldsymbol{\epsilon}_T. \tag{24}$$

The optimization of this noise calibration is inherently driven by the KD objective. This is predicated on the fact that achieving the lowest discrepancy between the denoised student and the teacher requires the student to be precisely aligned with the target noise intensity during the denoising phase.

### A.3. Locality-Sensitive Hashing (LSH) Guided Global Distillation

Figure 5 visualizes the mechanism of $\mathcal{L}_{\text{Global}}$. Within our framework, each hash function functions as a linear boundary. The aggregation of these boundaries partitions the feature space into numerous compact polyhedral regions. The primary

objective of the LSH loss is to constrain both $\boldsymbol{v}^{(stu)}$ and $\hat{\boldsymbol{v}}^{(stu)}$ to reside within the identical polyhedron. Consequently, increasing the number of hash functions effectively shrinks the volume of these polyhedra, thereby strictly bounding the angular deviation between the two vectors by $\eta$ (as shown in Figure 5). Crucially, since the polyhedron is narrow relative to the feature norms, this mechanism enforces directional alignment while decoupling the constraints on feature magnitudes.

The fundamental objective of LSH is to map spatially proximal points into identical buckets using a set of hash functions, thereby maximizing the collision probability for similar items. In our proposed framework, we employ a hash family constructed from the Gaussian distribution—known as a 2-stable distribution—which is formulated as:

$$\delta_{\boldsymbol{w},b}(\boldsymbol{v}) = \left\lfloor \frac{\boldsymbol{w}^\mathsf{T}\boldsymbol{v} + b}{r} \right\rfloor, \tag{25}$$

Here, $\boldsymbol{v} \in \mathbb{R}^D$ denotes the input feature vector, and $\boldsymbol{w} \in \mathbb{R}^D$ represents a random projection vector with entries drawn from a Gaussian distribution. The scalar $b$ is uniformly sampled from the interval $[0, r]$, where $r$ specifies the bin width. The notation $\lfloor \cdot \rfloor$ refers to the floor operator.

The proposed $\mathcal{L}_{\text{Global}}$ compels the student feature to reside in the identical bucket as the teacher. Drawing from the fundamental principles of LSH, the likelihood of collision exhibits a monotonic decay as the Euclidean distance between two vectors increases. Consequently, achieving hash consistency (i.e., $\delta_{\boldsymbol{W},b}(\hat{\boldsymbol{v}}^{(stu)}) = \delta_{\boldsymbol{W},b}(\boldsymbol{v}^{(stu)})$) serves as a prerequisite for feature identity ($\|\hat{\boldsymbol{v}}^{(stu)} - \boldsymbol{v}^{(stu)}\|_2 = 0$). This logical connection validates our strategy of enforcing feature mimicry via the minimization of $\mathcal{L}_{\text{Global}}$.

In the proposed framework, we employ a set of $M$ hash functions as formulated in Equation 25, producing an $M$-dimensional hash code for each input feature. By setting the threshold at zero to binarize the continuous values, the entire LSH module is operationally equivalent to a linear layer followed by a signum function:

$$\delta_{\boldsymbol{W},\boldsymbol{b}}(\boldsymbol{v}) = \text{sign}(\boldsymbol{W}^\mathsf{T}\boldsymbol{v} + \boldsymbol{b}), \tag{26}$$

$$\text{sign}(x) = \begin{cases} 1, & \text{if } x > 0; \\ 0, & \text{otherwise}, \end{cases} \tag{27}$$

Here, $\boldsymbol{v} \in \mathbb{R}^D$ denotes the feature vector, $\boldsymbol{W} \in \mathbb{R}^{D \times M}$ represents the weight matrix with entries drawn from a Gaussian distribution, and $\boldsymbol{b} \in \mathbb{R}^M$ indicates the bias term. Equation 26 yields $M$ binary codes for the teacher feature $\hat{\boldsymbol{v}}^{(stu)}$. To enforce alignment between the student's hash codes and the teacher's, we cast this objective as the minimization of a binary cross-entropy classification loss:

$$\mathcal{L}_{\text{Global}} = -\frac{1}{M} \sum_{m=1}^{M} \left[ \delta_m \log \rho_m + (1 - \delta_m) \log(1 - \rho_m) \right], \tag{28}$$

where $\boldsymbol{\delta} = \text{sign}(\boldsymbol{W}^\top \hat{\boldsymbol{v}}^{(stu)} + \boldsymbol{b}) \in \mathbb{R}^M$, $\boldsymbol{\rho} = \text{Sigmoid}(\boldsymbol{W}^\top \boldsymbol{v}^{(stu)} + \boldsymbol{b}) \in \mathbb{R}^M$, $\boldsymbol{W} \in \mathbb{R}^{D \times M}$ represents projection matrix whose values are sampled from a Gaussian distribution, and $\boldsymbol{b} \in \mathbb{R}^M$ denotes the bias. LSH relaxes the constraints of feature distillation on magnitude, but prioritize the alignment of feature direction.

### A.4. Theoretical Analyses of LSH Guided Global Distillation

We begin by investigating why $\mathcal{L}_{\text{Global}}$ depends on feature direction rather than magnitude. Specifically, Proposition A.1 demonstrates that $\mathcal{L}_{\text{Global}}$ remains constant even if the teacher features are scaled, indicating its invariance to the teacher's feature magnitude. To easy notation, we express the denoised student feature as the teacher role in this section.

**Proposition A.1.** *For a given scale $\zeta > 0$, $\mathcal{L}_{\text{Global}}(\boldsymbol{v}^{(stu)}, \zeta\hat{\boldsymbol{v}}^{(stu)}) = \mathcal{L}_{\text{Global}}(\boldsymbol{v}^{(stu)}, \hat{\boldsymbol{v}}^{(stu)})$ for arbitrary $\boldsymbol{v}^{(stu)}$.*

Subsequently, Proposition A.2 demonstrates that once the directions of $\boldsymbol{v}^{(stu)}$ and $\hat{\boldsymbol{v}}^{(stu)}$ are aligned, $\mathcal{L}_{\text{Global}}$ promotes an increase in the magnitude of $\boldsymbol{v}^{(stu)}$.

**Proposition A.2.** *Assume the direction of $\boldsymbol{v}^{(stu)}$ is the same as that of $\hat{\boldsymbol{v}}^{(stu)}$, and $\boldsymbol{b} = \boldsymbol{0}$ in LSH. For a given scale $\zeta > 1$, then $\mathcal{L}_{\text{Global}}(\zeta\boldsymbol{v}^{(stu)}, \hat{\boldsymbol{v}}^{(stu)}) \leq \mathcal{L}_{\text{Global}}(\boldsymbol{v}^{(stu)}, \hat{\boldsymbol{v}}^{(stu)})$ always holds.*

Crucially, Proposition A.3 and Proposition A.4 constitute our key theoretical insights, elucidating the mechanism by which the LSH loss enforces directional alignment between the student and teacher. Specifically, Proposition A.3 quantifies the probability of the loss falling below $\log 2$ conditioned on the angular discrepancy $\angle(\boldsymbol{v}^{(stu)}, \hat{\boldsymbol{v}}^{(stu)})$. Consequently, it demonstrates that a narrower angle directly correlates with a higher likelihood of achieving a minimal loss value.

Proposition A.4 quantifies the probability that the angular deviation $\angle(\boldsymbol{v}^{(stu)}, \hat{\boldsymbol{v}}^{(stu)})$ falls below $\epsilon$, conditioned on a minimized $\mathcal{L}_{\text{Global}}$. By numerically evaluating the cumulative probability based on this formulation, we observe that increasing the number of hashing functions directly enhances the likelihood of the student feature converging to the teacher's direction.

**Proposition A.3.** *Suppose $\boldsymbol{b} = \boldsymbol{0}$ in LSH, and $\boldsymbol{v}^{(stu)}$ and $\hat{\boldsymbol{v}}^{(stu)}$ follow the standard Gaussian distribution. Then,*

$$\Pr\{l_m < \log 2 \mid \angle(\boldsymbol{v}^{(stu)}, \hat{\boldsymbol{v}}^{(stu)}) = \eta\} = 1 - \frac{\eta}{\pi} \tag{29}$$

*will hold, where $\angle(\boldsymbol{v}^{(stu)}, \hat{\boldsymbol{v}}^{(stu)})$ denotes the angle between $\boldsymbol{v}^{(stu)}$ and $\hat{\boldsymbol{v}}^{(stu)}$, and*

$$l_m \doteq -\delta_m \log\left(\rho_m\right) - (1 - \delta_m) \log\left(1 - \rho_m\right). \tag{30}$$

**Proposition A.4.** *Suppose $\boldsymbol{b} = \boldsymbol{0}$ in LSH, and $\boldsymbol{v}^{(stu)}$ and $\hat{\boldsymbol{v}}^{(stu)}$ follow the standard Gaussian distribution. Then, for any $0 < \epsilon < \pi$, the equation*

$$\Pr\left\{\angle(\boldsymbol{v}^{(stu)}, \hat{\boldsymbol{v}}^{(stu)}) < \epsilon \mid \bigwedge_{m=1}^{M} (l_m < \log 2)\right\}$$
$$= \frac{\int_0^\epsilon \left(\left(1 - \frac{\eta}{\pi}\right)^N \cdot \sin^{D-2}\left(\eta\right)\right) \mathrm{d}\eta}{\int_0^\pi \left(\left(1 - \frac{\eta}{\pi}\right)^N \cdot \sin^{D-2}\left(\eta\right)\right) \mathrm{d}\eta} \tag{31}$$

*will hold.*

Based on these theoretical results, we perform a numerical evaluation of the probability in Proposition A.4 across varying values of $M$. We found that an increase in the number of hash functions $M$ leads the angle between student and teacher features to converge towards zero. This empirically validates that our LSH guided global distillation loss is highly effective in enforcing directional alignment.

We then show theoretical proofs of Proposition A.1-A.4.

### A.5. Proof of Proposition A.1

*Proof.* We will prove that $\delta_{\boldsymbol{W}, \mathbf{b}'}(\zeta \hat{\boldsymbol{v}}^{(stu)}) = \delta_{\boldsymbol{W}, \boldsymbol{b}}(\hat{\boldsymbol{v}}^{(stu)})$, where $\boldsymbol{W} = [\boldsymbol{w}_1, \boldsymbol{w}_2, \cdots, \boldsymbol{w}_M]^\mathsf{T}$, $\mathbf{b}' = [b_1', b_2', \cdots, b_M']^\mathsf{T}$ and $\boldsymbol{b} = [b_1, b_2, \cdots, b_M]^\mathsf{T}$. Assume there are $\ell$ teacher features, that is $\hat{\boldsymbol{v}}_1^{(stu)}, \hat{\boldsymbol{v}}_2^{(stu)}, \cdots, \hat{\boldsymbol{v}}_\ell^{(stu)}$.

For $0 \leq m \leq M$, $b_m = \text{median}(\hat{\boldsymbol{v}}_1^{(stu)}, \hat{\boldsymbol{v}}_2^{(stu)}, \cdots, \hat{\boldsymbol{v}}_\ell^{(stu)})$. And it is easy to see $b_m' = \text{median}(\zeta \hat{\boldsymbol{v}}_1^{(stu)}, \zeta \hat{\boldsymbol{v}}_2^{(stu)}, \cdots, \zeta \hat{\boldsymbol{v}}_\ell^{(stu)}) = \zeta b_m$ when $\zeta > 0$. Hence, $\text{sign}(\boldsymbol{w}_m^\mathsf{T} \hat{\boldsymbol{v}}_i^{(stu)} + b_m) = \text{sign}(\boldsymbol{w}_m^\mathsf{T} \zeta \hat{\boldsymbol{v}}_i^{(stu)} + \zeta b_m)$ for $0 \leq i \leq \ell$ when $\zeta > 0$. That implies $\delta_{\boldsymbol{W}, \mathbf{b}'}(\zeta \hat{\boldsymbol{v}}^{(stu)}) = \delta_{\boldsymbol{W}, \boldsymbol{b}}(\hat{\boldsymbol{v}}^{(stu)})$. $\square$

### A.6. Proof of Proposition A.2

*Proof.* Note that

$$\mathcal{L}_{\text{Global}} = -\frac{1}{M} \sum_{m=1}^{M} \left[\delta_m \log \rho_m + (1 - \delta_m) \log(1 - \rho_m)\right], \tag{32}$$

where $\delta_m$ and $\rho_m$ is the $m$-th entry of $\text{sign}(\boldsymbol{W}^\mathsf{T} \hat{\boldsymbol{v}}^{(stu)})$ and $\text{Sigmoid}(\boldsymbol{W}^\mathsf{T} \boldsymbol{v}^{(stu)})$, respectively.

We begin by examining the case where $\delta_m = 1$, which implies that $\boldsymbol{w}_m$ and $\hat{\boldsymbol{v}}^{(stu)}$ form an acute angle, thereby ensuring

$\cos \langle \boldsymbol{w}_m, \hat{\boldsymbol{v}}^{(stu)} \rangle \geq 0$. Therefore,

$$-\delta_m \log \text{sigmoid}(\zeta \boldsymbol{w}_m^\mathsf{T} \boldsymbol{v}^{(stu)}) - (1 - \delta_m) \log(1 - \text{sigmoid}(\zeta \boldsymbol{w}_m^\mathsf{T} \boldsymbol{v}^{(stu)}))$$

$$= -\log \text{sigmoid}(\zeta \|\boldsymbol{w}_m\| \|\boldsymbol{v}^{(stu)}\| \cos \langle \boldsymbol{w}_m, \boldsymbol{v}^{(stu)} \rangle) \tag{33}$$

$$\leq -\log \text{sigmoid}(\|\boldsymbol{w}_m\| \|\boldsymbol{v}^{(stu)}\| \cos \langle \boldsymbol{w}_m, \hat{\boldsymbol{v}}^{(stu)} \rangle) \tag{34}$$

$$= -\delta_m \log \rho_m - (1 - \delta_m) \log(1 - \rho_m). \tag{35}$$

Then, when $\delta_m = 0$, similar to equation 33, we can get

$$-\delta_m \log \text{sigmoid}(\zeta \boldsymbol{w}_m^\mathsf{T} \boldsymbol{v}^{(stu)}) - (1 - \delta_m) \log(1 - \text{sigmoid}(\zeta \boldsymbol{w}_m^\mathsf{T} \boldsymbol{v}^{(stu)}))$$

$$= -\log \left( 1 - \text{sigmoid}(\zeta \boldsymbol{w}_m^\mathsf{T} \boldsymbol{v}^{(stu)}) \right) \tag{36}$$

$$\leq -\log \left( 1 - \text{sigmoid}(\|\boldsymbol{w}_m\| \|\boldsymbol{v}^{(stu)}\| \cos \langle \boldsymbol{w}_m, \hat{\boldsymbol{v}}^{(stu)} \rangle) \right) \tag{37}$$

$$= -\delta_m \log \rho_m - (1 - \delta_m) \log(1 - \rho_m). \tag{38}$$

In summary, $\mathcal{L}_{\text{Global}}(\zeta \boldsymbol{v}^{(stu)}, \hat{\boldsymbol{v}}^{(stu)}) \leq \mathcal{L}_{\text{Global}}(\boldsymbol{v}^{(stu)}, \hat{\boldsymbol{v}}^{(stu)})$ always holds when $\zeta > 1$. $\qquad \square$

### A.7. Proof of Proposition A.3 and A.4

We begin by establishing the notations and definitions used in our analysis. Specifically, we postulate that both $\boldsymbol{v}^{(stu)}$ and $\hat{\boldsymbol{v}}^{(stu)}$ are drawn from a standard normal distribution:

- $\hat{\boldsymbol{v}}^{(stu)} \in \mathbb{R}^D : \hat{\boldsymbol{v}}^{(stu)} \sim \mathcal{N}(\mathbf{0}, \mathbf{I}_D)$

- $\boldsymbol{v}^{(stu)} \in \mathbb{R}^D : \boldsymbol{v}^{(stu)} \sim \mathcal{N}(\mathbf{0}, \mathbf{I}_D)$

Within the LSH framework, we represent $\boldsymbol{W} \in \mathbb{R}^{D \times M}$ as the stack of vectors $[\boldsymbol{w}_1, \boldsymbol{w}_2, \cdots, \boldsymbol{w}_M]^\mathsf{T}$, where all elements are initialized via Gaussian sampling:

- $\boldsymbol{w}_m \in \mathbb{R}^D : \boldsymbol{w}_m \sim \mathcal{N}(\mathbf{0}, \mathbf{I}_D)$

A few derived variables are:

- $\boldsymbol{\delta} \in \mathbb{R}^M : \delta_m \doteq \text{sign}\left(\boldsymbol{w}_m^\mathsf{T} \hat{\boldsymbol{v}}^{(stu)}\right)$

- $\boldsymbol{\rho} \in \mathbb{R}^M : \rho_m \doteq \sigma\left(\boldsymbol{w}_m^\mathsf{T} \boldsymbol{v}^{(stu)}\right)$

- $\boldsymbol{l} \in \mathbb{R}^M : l_m \doteq -\delta_m \log\left(\rho_m\right) - (1 - \delta_m) \log\left(1 - \rho_m\right)$

We also use a few shorthand notations:

- $U(\boldsymbol{x}) \doteq \begin{cases} \frac{\boldsymbol{x}}{\|\boldsymbol{x}\|_2} & \text{if } \|\boldsymbol{x}\|_2 > 0, \\ \mathbf{0} & \text{otherwise}. \end{cases}$

- $\angle(\boldsymbol{x}, \boldsymbol{y}) \doteq \arccos\left(U(\boldsymbol{x})^\mathsf{T} U(\boldsymbol{y})\right)$

- $\mathbb{X}^{n-1} \doteq \{\boldsymbol{x} \in \mathbb{R}^n \mid \|\boldsymbol{x}\|_2 = 1\}$: the $n$-dimensional unit hypersphere

- $\omega$: the geodesic distance, with which $\mathbb{X}^{n-1}$ forms a legitimate metric space

- $\mu_n$: the Lebesgue measure on $\mathbb{R}^n$

- $\sigma_{n-1}$: the surface area measure on $\mathbb{X}^{n-1}$

- $S_{n-1} \doteq \int_{\mathbb{X}^{n-1}} \mathrm{d}\sigma_{n-1} = \frac{2\pi^{n/2}}{\Gamma(n/2)}$: the surface area of $\mathbb{X}^{n-1}$

- $\mathbb{I}$: the indicator function

- $p(x)$: the p.d.f. of $x$

**Lemma 1.** *Let $\boldsymbol{x} \in \mathbb{R}^n$ ($n \in \mathbb{N}^+$) be a random vector with each element $x_i \sim \mathcal{N}(0, 1)$ independently. Then for any function $f : \mathbb{X}^{n-1} \cup \{\boldsymbol{0}\} \to \mathbb{R}$ satisfying*

- *$f$ is bounded,*

- *$f$ is continuous on $\mathbb{X}^{n-1}$,*

*there holds*

$$\mathbb{E}_{\boldsymbol{x}}[f(U(\boldsymbol{x}))] = \frac{1}{S_{n-1}} \int_{\mathbb{X}^{n-1}} f(\boldsymbol{u}) \, \mathrm{d}\sigma_{n-1}(\boldsymbol{u}) . \tag{39}$$

*Proof.*

$$\mathbb{E}_{\boldsymbol{x}}[f(U(\boldsymbol{x}))]$$

$$= \int_{\mathbb{R}^n} f(U(\boldsymbol{x})) \cdot p(\boldsymbol{x}) \, \mathrm{d}\mu_n(\boldsymbol{x}) +$$
$$\int_{\{\boldsymbol{0}\}} f(U(\boldsymbol{x})) \cdot p(\boldsymbol{x}) \, \mathrm{d}\mu_n(\boldsymbol{x}) \tag{40}$$

$$= \int_{\mathbb{R}^n} f(U(\boldsymbol{x})) \cdot p(\boldsymbol{x}) \, \mathrm{d}\mu_n(\boldsymbol{x}) + \underbrace{0}_{\text{due to } f\text{'s boundedness}} \tag{41}$$

$$= \int_{\mathbb{R}^n} f(U(\boldsymbol{x})) \cdot (2\pi)^{-n/2} e^{-\frac{1}{2}\|\boldsymbol{x}\|_2^2} \, \mathrm{d}\mu_n(\boldsymbol{x}) \tag{42}$$

$$= (2\pi)^{-n/2} \underbrace{\int_0^\infty \left[ \int_{\mathbb{X}^{n-1}} f(\boldsymbol{u}) \, \mathrm{d}\sigma_{n-1}(\boldsymbol{u}) \right] \cdot e^{-\frac{r^2}{2}} r^{n-1} \, \mathrm{d}r}_{\text{integration by substitution (from Cartesian to polar)}} \tag{43}$$

$$= (2\pi)^{-n/2} \int_0^\infty \left( e^{-r^2/2} r^{n-1} \right) \mathrm{d}r \int_{\mathbb{X}^{n-1}} f(\boldsymbol{u}) \, \mathrm{d}\sigma_{n-1}(\boldsymbol{u}) \tag{44}$$

$$= \underbrace{\frac{\Gamma(n/2)}{2\pi^{n/2}}}_{\frac{1}{S_{n-1}}} \int_{\mathbb{X}^{n-1}} f(\boldsymbol{u}) \, \mathrm{d}\sigma_{n-1}(\boldsymbol{u}) . \tag{45}$$

$\square$

**Corollary 1.1.** *Let $\boldsymbol{x} \in \mathbb{R}^n$ ($n \in \mathbb{N}^+$) be a random vector with each element $x_i \sim \mathcal{N}(0, 1)$ independently. Then for any Borel set $\mathbb{B}$ in $\mathbb{X}^{n-1}$,*

$$\Pr\{U(\boldsymbol{x}) \in \mathbb{B}\} = \frac{\sigma_{n-1}(\mathbb{B})}{S_{n-1}} . \tag{46}$$

*Proof.* For an open set $\mathbb{O}$ in $\mathbb{X}^{n-1}$ (rename it to make things clear), define

$$f_{\mathbb{O}}(\boldsymbol{u}) = \begin{cases} 0, & \text{for } \boldsymbol{u} = \boldsymbol{0}; \\ \chi_{\mathbb{O}}, & \text{for } \boldsymbol{u} \in \mathbb{X}^{n-1} , \end{cases} \tag{47}$$

where $\chi_{\mathbb{O}}$ is the characteristic function of $\mathbb{O}$ and

$$f_{\mathbb{O}}^{(k)}(\boldsymbol{u}) = \begin{cases} 0 & \text{if } \boldsymbol{u} = \boldsymbol{0}; \\ \max\left(0, 1 - k \cdot \inf_{\boldsymbol{v} \in \mathbb{O}} \omega(\boldsymbol{u}, \boldsymbol{v})\right) & \text{if } \boldsymbol{u} \in \mathbb{X}^{n-1} \end{cases} \tag{48}$$

for $k \in \mathbb{N}^*$. Then the conditions of Lebesgue's dominated convergence theorem are met, i.e.,

- $f_{\mathbb{O}}^{(k)}$'s are bounded,

- $f_{\mathbb{O}}^{(k)}$'s converge pointwise to $f_{\mathbb{O}}$.

Thus

$$\Pr\{U(\boldsymbol{x}) \in \mathbb{O}\}$$

$$= \mathbb{E}_{\boldsymbol{x}}[f_{\mathbb{O}}(U(\boldsymbol{x}))] \tag{49}$$

$$= \underbrace{\lim_{k \to \infty} \mathbb{E}_{\boldsymbol{x}}\left[f_{\mathbb{O}}^{(k)}(U(\boldsymbol{x}))\right]}_{\text{due to dominated convergence theorem}} \tag{50}$$

$$= \lim_{k \to \infty}\left(\frac{1}{S_{n-1}}\int_{\mathbb{X}^{n-1}} f_{\mathbb{O}}^{(k)}(\boldsymbol{u})\,\mathrm{d}\sigma_{n-1}(\boldsymbol{u})\right) \tag{51}$$

$$= \frac{1}{S_{n-1}}\lim_{k \to \infty}\left(\int_{\mathbb{X}^{n-1}} f_{\mathbb{O}}^{(k)}(\boldsymbol{u})\,\mathrm{d}\sigma_{n-1}(\boldsymbol{u})\right) \tag{52}$$

$$= \frac{1}{S_{n-1}}\underbrace{\int_{\mathbb{X}^{n-1}} f_{\mathbb{O}}(\boldsymbol{u})\,\mathrm{d}\sigma_{n-1}(\boldsymbol{u})}_{\text{due to dominated convergence theorem}} \tag{53}$$

$$= \frac{1}{S_{n-1}}\int_{\mathbb{X}^{n-1}} \chi_{\mathbb{O}}(\boldsymbol{u})\,\mathrm{d}\sigma_{n-1}(\boldsymbol{u}) \tag{54}$$

$$= \frac{1}{S_{n-1}}\int_{\mathbb{O}} \mathrm{d}\sigma_{n-1}(\boldsymbol{u}) \tag{55}$$

$$= \frac{\sigma_{n-1}(\mathbb{O})}{S_{n-1}}. \tag{56}$$

Now that the equation holds for any open set $\mathbb{O}$, it can be shown by induction that

$$\Pr\{U(\boldsymbol{x}) \in \mathbb{B}\} = \frac{\sigma_{n-1}(\mathbb{B})}{S_{n-1}} \tag{57}$$

for any Borel set $\mathbb{B}$. □

**Lemma 2.**

$$p\left\{\angle(\hat{\boldsymbol{v}}^{(stu)}, \boldsymbol{v}^{(stu)}) = \eta\right\} = \frac{S_{D-2}}{S_{D-1}}\sin^{D-2}(\eta) \tag{58}$$

*for $\eta \in (0, \pi)$.*

*Proof.*

$$\Pr\left\{\angle(\hat{\boldsymbol{v}}^{(stu)}, \boldsymbol{v}^{(stu)}) \le \eta \mid \hat{\boldsymbol{v}}^{(stu)}\right\}$$

$$= \Pr\left\{U(\boldsymbol{v}^{(stu)}) \in \{\boldsymbol{x} \in \mathbb{X}^{D-1} \mid \omega(U(\hat{\boldsymbol{v}}^{(stu)}), \boldsymbol{x}) \le \eta\} \mid \hat{\boldsymbol{v}}^{(stu)}\right\} \tag{59}$$

$$= \frac{1}{S_{D-1}}\sigma_{D-1}\left(\{\boldsymbol{x} \in \mathbb{X}^{D-1} \mid \omega(U(\hat{\boldsymbol{v}}^{(stu)}), \boldsymbol{x}) \le \eta\}\right) \tag{60}$$

$$= \frac{1}{S_{D-1}}\int_0^\eta \left(S_{D-2}\sin^{D-2}(\phi)\right)\mathrm{d}\phi \tag{61}$$

$$= \frac{S_{D-2}}{S_{D-1}}\int_0^\eta \sin^{D-2}(\phi)\,\mathrm{d}\phi. \tag{62}$$

Then

$$\Pr\left\{\angle(\hat{\boldsymbol{v}}^{(stu)}, \boldsymbol{v}^{(stu)}) \le \eta\right\}$$

$$= \int_{\mathbb{R}^D} \left(\Pr\left\{\angle(\hat{\boldsymbol{v}}^{(stu)}, \boldsymbol{v}^{(stu)}) \le \eta \mid \hat{\boldsymbol{v}}^{(stu)}\right\} \cdot p(\hat{\boldsymbol{v}}^{(stu)})\right) \, \mathrm{d}\mu_D(\hat{\boldsymbol{v}}^{(stu)}) \tag{63}$$

$$= \int_{\mathbb{R}^D} \left(\left(\frac{S_{D-2}}{S_{D-1}} \int_0^\eta \sin^{D-2}(\phi) \, \mathrm{d}\phi\right) \cdot p(\hat{\boldsymbol{v}}^{(stu)})\right) \, \mathrm{d}\mu_D(\hat{\boldsymbol{v}}^{(stu)}) \tag{64}$$

$$= \left(\int_{\mathbb{R}^D} \left(p(\hat{\boldsymbol{v}}^{(stu)})\right) \, \mathrm{d}\mu_D(\hat{\boldsymbol{v}}^{(stu)})\right) \left(\frac{S_{D-2}}{S_{D-1}} \int_0^\eta \sin^{D-2}(\phi) \, \mathrm{d}\phi\right) \tag{65}$$

$$= \frac{S_{D-2}}{S_{D-1}} \int_0^\eta \sin^{D-2}(\phi) \, \mathrm{d}\phi \,. \tag{66}$$

Thus

$$p\left\{\angle(\hat{\boldsymbol{v}}^{(stu)}, \boldsymbol{v}^{(stu)}) = \eta\right\} = \frac{\mathrm{d}}{\mathrm{d}\eta} \Pr\left\{\angle(\hat{\boldsymbol{v}}^{(stu)}, \boldsymbol{v}^{(stu)}) \le \eta\right\} \tag{67}$$

$$= \frac{\mathrm{d}}{\mathrm{d}\eta} \left(\frac{S_{D-2}}{S_{D-1}} \int_0^\eta \sin^{D-2}(\phi) \, \mathrm{d}\phi\right) \tag{68}$$

$$= \frac{S_{D-2}}{S_{D-1}} \frac{\mathrm{d}}{\mathrm{d}\eta} \int_0^\eta \sin^{D-2}(\phi) \, \mathrm{d}\phi \tag{69}$$

$$= \frac{S_{D-2}}{S_{D-1}} \sin^{D-2}(\eta) \,. \tag{70}$$

$\square$

### A.7.1. PROOF OF PROPOSITION A.3

*Proof.* First, let us inspect the properties of the $l_m$'s, which can be rewritten as

$$l_m = \begin{cases} -\log(\rho_m) & \text{if } \delta_m = 1 \,, \\ -\log(1 - \rho_m) & \text{if } \delta_m = 0 \,. \end{cases} \tag{71}$$

Note that $\log 2 = -\log\left(1 - \frac{1}{2}\right)$. Thus $l_m < \log 2$ if and only if

$$\boldsymbol{\rho}[j] \begin{cases} > \frac{1}{2} & \text{if } \boldsymbol{\delta}[j] = 1 \,, \\ < \frac{1}{2} & \text{if } \boldsymbol{\delta}[j] = 0 \,, \end{cases} \tag{72}$$

which is equivalent to

$$\boldsymbol{w}_m^\mathsf{T} \boldsymbol{v}^{(stu)} \begin{cases} > 0 & \text{if } \boldsymbol{w}_m^\mathsf{T} \hat{\boldsymbol{v}}^{(stu)} > 0 \,, \\ < 0 & \text{if } \boldsymbol{w}_m^\mathsf{T} \hat{\boldsymbol{v}}^{(stu)} \le 0 \,. \end{cases} \tag{73}$$

In other words,

$$U(\boldsymbol{w}_m) \in \mathbb{L}_{\hat{\boldsymbol{v}}^{(stu)}, \boldsymbol{v}^{(stu)}} \,, \tag{74}$$

where

$$\mathbb{L}_{\hat{\boldsymbol{v}}^{(stu)}, \boldsymbol{v}^{(stu)}} = \left\{\boldsymbol{x} \in \mathbb{X}^{D-1} \mid \left(\left(\boldsymbol{x}^\mathsf{T} \boldsymbol{v}^{(stu)} > 0\right) \wedge \left(\boldsymbol{x}^\mathsf{T} \hat{\boldsymbol{v}}^{(stu)} > 0\right)\right)\right\} \vee$$
$$\left\{\boldsymbol{x} \in \mathbb{X}^{D-1} \mid \left(\left(\boldsymbol{x}^\mathsf{T} \boldsymbol{v}^{(stu)} < 0\right) \wedge \left(\boldsymbol{x}^\mathsf{T} \hat{\boldsymbol{v}}^{(stu)} \le 0\right)\right)\right\} \tag{75}$$

is the union of two lunes.

Applying Corollary 1.1, we have

$$\Pr\left\{U(\boldsymbol{w}_m) \in \mathbb{L}_{\hat{\boldsymbol{v}}^{(stu)}, \boldsymbol{v}^{(stu)}} \mid \hat{\boldsymbol{v}}^{(stu)}, \boldsymbol{v}^{(stu)}\right\}$$

$$=\frac{\sigma_{D-1}\left(\mathbb{L}_{\hat{\boldsymbol{v}}^{(stu)}, \boldsymbol{v}^{(stu)}}\right)}{S_{D-1}} \tag{76}$$

$$=\frac{2}{S_{D-1}} \int_0^{\frac{\pi}{2}} \underbrace{\frac{\pi - \angle\left(\hat{\boldsymbol{v}}^{(stu)}, \boldsymbol{v}^{(stu)}\right)}{2\pi} A_1 \cos(\eta) A_{D-3} \sin^{D-3}(\eta) \mathrm{d}\eta}_{\mathbb{X}^{D-1} \text{ viewed as a union of tori}} \tag{77}$$

$$=\frac{\pi - \angle\left(\hat{\boldsymbol{v}}^{(stu)}, \boldsymbol{v}^{(stu)}\right)}{\pi S_{D-1}} \int_0^{\frac{\pi}{2}} \left(A_1 \cos(\eta) \cdot A_{D-3} \sin^{D-3}(\eta)\right) \mathrm{d}\eta \tag{78}$$

$$=\frac{\pi - \angle\left(\hat{\boldsymbol{v}}^{(stu)}, \boldsymbol{v}^{(stu)}\right)}{\pi S_{D-1}} S_{D-1} \tag{79}$$

$$=1 - \frac{\angle\left(\hat{\boldsymbol{v}}^{(stu)}, \boldsymbol{v}^{(stu)}\right)}{\pi}. \tag{80}$$

Thus

$$\Pr\left\{U(\boldsymbol{w}_m) \in \mathbb{L}_{\hat{\boldsymbol{v}}^{(stu)}, \boldsymbol{v}^{(stu)}} \mid \angle(\hat{\boldsymbol{v}}^{(stu)}, \boldsymbol{v}^{(stu)}) = \eta\right\}$$

$$=\int_{\mathbb{R}^{D \times D}} (\Pr\left\{U(\boldsymbol{w}_m) \in \mathbb{L}_{\hat{\boldsymbol{v}}^{(stu)}, \boldsymbol{v}^{(stu)}} \mid \hat{\boldsymbol{v}}^{(stu)}, \boldsymbol{v}^{(stu)}\right\}\cdot$$

$$p\left(\hat{\boldsymbol{v}}^{(stu)}, \boldsymbol{v}^{(stu)} \mid \angle(\hat{\boldsymbol{v}}^{(stu)}, \boldsymbol{v}^{(stu)}) = \eta\right))\mathrm{d}\mu_{D \times D}(\hat{\boldsymbol{v}}^{(stu)}, \boldsymbol{v}^{(stu)}) \tag{81}$$

$$=\int_{\mathbb{R}^{D \times D}} \left(1 - \frac{\eta}{\pi}\right) \cdot p\left(\hat{\boldsymbol{v}}^{(stu)}, \boldsymbol{v}^{(stu)} \mid \angle(\hat{\boldsymbol{v}}^{(stu)}, \boldsymbol{v}^{(stu)}) = \eta\right)\mathrm{d}\mu_{D \times D}(\hat{\boldsymbol{v}}^{(stu)}, \boldsymbol{v}^{(stu)}) \tag{82}$$

$$=\left(1 - \frac{\eta}{\pi}\right) \int_{\mathbb{R}^{D \times D}} p\left(\hat{\boldsymbol{v}}^{(stu)}, \boldsymbol{v}^{(stu)} \mid \angle(\hat{\boldsymbol{v}}^{(stu)}, \boldsymbol{v}^{(stu)}) = \eta\right) \mathrm{d}\mu_{D \times D}(\hat{\boldsymbol{v}}^{(stu)}, \boldsymbol{v}^{(stu)}) \tag{83}$$

$$=1 - \frac{\eta}{\pi}. \tag{84}$$

$$\square$$

### A.7.2. PROOF OF PROPOSITION A.4

*Proof.*

$$\Pr\left\{\bigwedge_{m=1}^{M} (l_m < \log 2) \mid \angle(\hat{\boldsymbol{v}}^{(stu)}, \boldsymbol{v}^{(stu)}) = \eta\right\}$$

$$=\underbrace{\prod_{j=1}^{M} \Pr\left\{l_m < \log 2 \mid \angle(\hat{\boldsymbol{v}}^{(stu)}, \boldsymbol{v}^{(stu)}) = \eta\right\}}_{\text{due to conditional independence of } l_m\text{'s}} \tag{85}$$

$$=\underbrace{\left(1 - \frac{\eta}{\pi}\right)^{M}}_{\text{due to Proposition A.3}}. \tag{86}$$

Applying the Bayes' rule, the conditional probability density of $\angle(\hat{\boldsymbol{v}}^{(stu)}, \boldsymbol{v}^{(stu)})$ can be derived as

$$
p\left\{\angle(\hat{\boldsymbol{v}}^{(stu)}, \boldsymbol{v}^{(stu)}) = \eta \mid \bigwedge_{m=1}^{M} (l_m < \log 2)\right\}
$$

$$
= \frac{p\left\{\bigwedge_{m=1}^{M}(l_m < \log 2) \mid \angle(\hat{\boldsymbol{v}}^{(stu)}, \boldsymbol{v}^{(stu)}) = \eta\right\} \cdot p\left\{\angle(\hat{\boldsymbol{v}}^{(stu)}, \boldsymbol{v}^{(stu)}) = \eta\right\}}{p\left\{\bigwedge_{m=1}^{M}(l_m < \log 2)\right\}} \tag{87}
$$

$$
= \frac{p\left\{\bigwedge_{m=1}^{M}(l_m < \log 2) \mid \angle(\hat{\boldsymbol{v}}^{(stu)}, \boldsymbol{v}^{(stu)}) = \eta\right\} p\left\{\angle(\hat{\boldsymbol{v}}^{(stu)}, \boldsymbol{v}^{(stu)}) = \eta\right\}}{\int_0^\pi p\left\{\bigwedge_{m=1}^{M}(l_m < \log 2) \mid \angle(\hat{\boldsymbol{v}}^{(stu)}, \boldsymbol{v}^{(stu)}) = \eta\right\} p\left\{\angle(\hat{\boldsymbol{v}}^{(stu)}, \boldsymbol{v}^{(stu)}) = \eta\right\} \mathrm{d}\eta} \tag{88}
$$

$$
= \frac{\left(1 - \frac{\eta}{\pi}\right)^M \cdot \frac{S_{D-2}}{S_{D-1}} \sin^{D-2}(\eta)}{\underbrace{\int_0^\pi \left(\left(1 - \frac{\eta}{\pi}\right)^M \cdot \frac{S_{D-2}}{S_{D-1}} \sin^{D-2}(\eta)\right) \mathrm{d}\eta}_{\text{due to Equation 85 and Corollary 2}}} \tag{89}
$$

$$
= \frac{\left(1 - \frac{\eta}{\pi}\right)^M \cdot \sin^{D-2}(\eta)}{\int_0^\pi \left(\left(1 - \frac{\eta}{\pi}\right)^M \cdot \sin^{D-2}(\eta)\right) \mathrm{d}\eta}. \tag{90}
$$

Thus

$$
\Pr\left\{\angle(\hat{\boldsymbol{v}}^{(stu)}, \boldsymbol{v}^{(stu)}) < \epsilon \mid \bigwedge_{m=1}^{M}(l_m < \log 2)\right\}
$$

$$
= \int_0^\epsilon p\left\{\angle(\hat{\boldsymbol{v}}^{(stu)}, \boldsymbol{v}^{(stu)}) = \eta \mid \bigwedge_{m=1}^{M}(l_m < \log 2)\right\} \mathrm{d}\eta \tag{91}
$$

$$
= \int_0^\epsilon \left(\frac{\left(1 - \frac{\eta}{\pi}\right)^M \cdot \sin^{D-2}(\eta)}{\int_0^\pi \left(\left(1 - \frac{\eta}{\pi}\right)^M \cdot \sin^{D-2}(\eta)\right) \mathrm{d}\eta}\right) \mathrm{d}\eta \tag{92}
$$

$$
= \frac{\int_0^\epsilon \left(\left(1 - \frac{\eta}{\pi}\right)^M \cdot \sin^{D-2}(\eta)\right) \mathrm{d}\eta}{\int_0^\pi \left(\left(1 - \frac{\eta}{\pi}\right)^M \cdot \sin^{D-2}(\eta)\right) \mathrm{d}\eta}. \tag{93}
$$

$\square$

## B. Experiments

### B.1. Experimental Setup

**Datasets.** We adopt CIFAR-100 (Krizhevsky & Hinton, 2009) and ImageNet (Deng et al., 2009) datasets for image classification. CIFAR-100 dataset consists of 50K training images and 10K testing images across 100 classes. ImageNet is a large-scale dataset composed of 1.2 million training images and 50K validation images in 1000 classes. COCO (Lin et al., 2014) contains 120k training images and 5k validation images. ADE20K (Zhou et al., 2017) contains 20K training images and 2K validation images with 150 semantic categories under diverse scenes for large-scale semantic segmentation.

*Table 9.* Training strategies on image classification tasks. **BS**: batch size; **LR**: learning rate; **WD**: weight decay.

| Strategy | Dataset | Epochs | Total BS | Initial LR | Optimizer | WD | LR scheduler | Data augmentation |
|---|---|---|---|---|---|---|---|---|
| A1 | CIFAR-100 | 240 | 64 | 0.05 | SGD | $5 \times 10^{-4}$ | $\times 0.1$ at 150,180,210 epochs | crop + flip |
| B1 | ImageNet | 100 | 256 | 0.1 | SGD | $1 \times 10^{-4}$ | $\times 0.1$ every 30 epochs | crop + flip |
| B2 | ImageNet | 300 | 1024 | 0.001 | AdamW | 0.05 | cosine decay | (Liu et al., 2021) |

## B.2. Analysis of The Number of Denoising Timesteps $T$

The number of denoising timesteps is an important factor for the denoising quality of student features. In Table 10, we investigate the impact on various numbers of denoising timesteps on CIFAR-100 and ImageNet. We observed that more timesteps generally produce better accuracy due to stronger denoising capability. However, more timesteps often generate additional training costs. And we found that $T = 2$ on CIFAR-100 and $T = 3$ on ImageNet are good enough for DSKD for pursuing the best efficiency-accuracy trade-off. We adopt the larger number of timesteps on ImageNet than that of timesteps on CIFAR-100, because the ImageNet's images have larger resolution sizes and more complex objects than CIFAR-100's images.

*Table 10.* Top-1 accuracy results under various numbers of denoising timesteps on CIFAR-100 and ImageNet.

| Timesteps $T$ | 1 | 2 | 3 | 4 | 5 |
|---|---|---|---|---|---|
| CIFAR-100 | 74.20 | 74.45 | 74.42 | 74.49 | 74.47 |
| ImageNet | 71.93 | 72.34 | 72.57 | 72.58 | 72.60 |

## B.3. Analysis of the Number of Hash Functions $M$ for LSH-Guided Feature Distillation

As shown in Table 11, we explore the number of hash functions $M$ for LSH-guided feature distillation. In theory, a larger $M$ reduces randomness of LSH for feature distillation. We observe that $M = 256$ achieves the best performance.

*Table 11.* Top-1 accuracy results under various numbers of hash functions $M$ on CIFAR-100 and ImageNet.

| Hash function number $M$ | 32 | 64 | 128 | 256 | 512 |
|---|---|---|---|---|---|
| CIFAR-100 | 74.28 | 74.36 | 74.39 | 74.45 | 74.44 |
| ImageNet | 72.38 | 72.49 | 72.51 | 72.57 | 72.55 |

## B.4. Analysis of Various Feature Distillation Mechanisms

We compare LSH-guided feature distillation mechanism with existing alternative feature distillation works like the traditional FitNet (Romero et al., 2014) and FNKD (Xu et al., 2020). As shown in Table 12, our LSE-guided feature distillation outperforms the existing popular feature distillation mechanisms.

*Table 12.* Top-1 accuracy results under various feature distillation mechanisms on CIFAR-100 and ImageNet.

| Mechanisms | FitNet | FNKD | LSH |
|---|---|---|---|
| CIFAR-100 | 73.94 | 74.08 | **74.45** |
| ImageNet | 72.16 | 72.28 | **72.57** |

## B.5. Analysis of Class Probability Distribution

As shown in Figure 6, we show the distributions of predicted class probabilities by ResNet-18 pretrained on ImageNet using teacher model, our DSKD, DiffKD, and the baseline student, respectively. Compared to DiffKD and baseline student, our DSKD leads to higher prediction quality: (1) Even in the case of an easy sample that all methods are correctly predicted, our DSKD assigns the highest probability to the correct class; (2) In the hard case, only DSKD makes correct predictions, while other methods produce error predictions. The results verify that our DSKD can guide the model to learn more similar distributions with the teacher.

## B.6. Visualization of attention maps.

As shown in Figure 7, we visualize Grad-CAM (Selvaraju et al., 2017) attention heatmaps generated by the teacher as well as the student trained by baseline, DiffKD, and our DSKD on ImageNet. We observed that our method learns more similar attention patterns and highlighted class-related features with the teacher. This enables the student to focus more precisely on

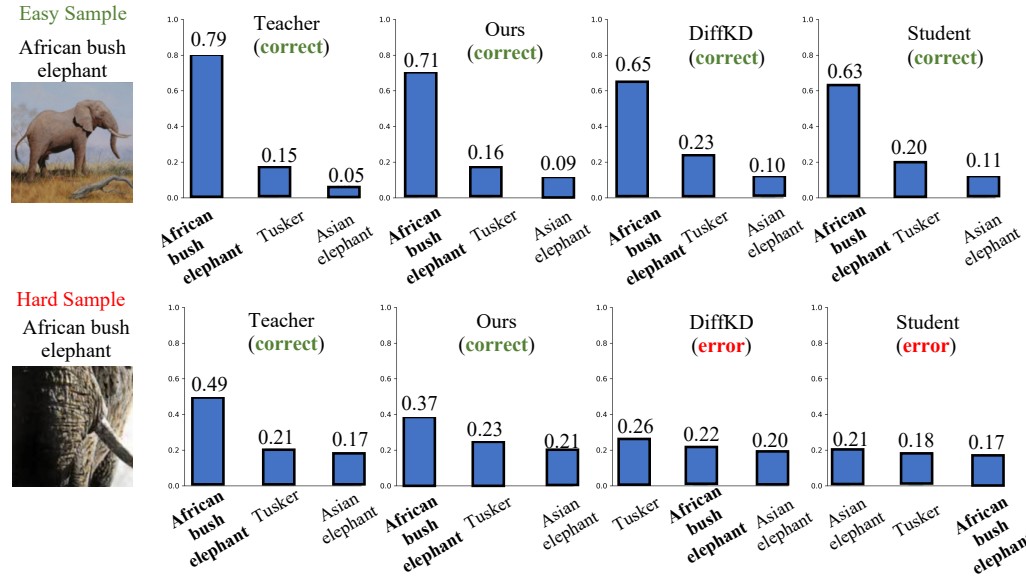

*Figure 6.* Illustrations of class probability distributions generated by ResNet-18 pretrained on ImageNet using teacher, our DSKD, DiffKD, and the baseline student, respectively.

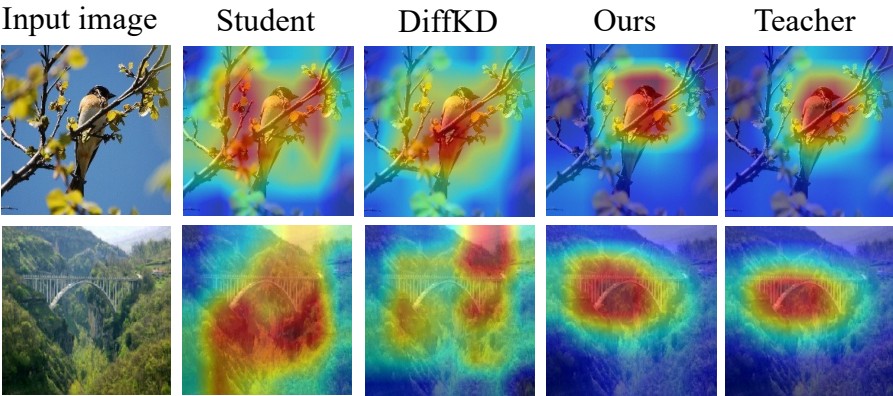

*Figure 7.* Visualizations of attention heatmaps generated by teacher as well as student trained by baseline, DiffKD and our DSKD on ImageNet.

target objects, further enhancing its performance. This can be attributed to *teacher-classifier-guided* diffusion model that improves the representational capability of student features.

### B.7. Results on Few-Shot Scenario

In the real-world scenario, the training samples are often scarce. Therefore, we further examine the effectiveness of DSKD under few-shot scenario. As shown in Fig.8, we chooses 25%, 50%, and 75% training samples from the original training set for distillation, while reserving the original test set. For a fair comparison, we adopt a class-balanced data split strategy and retain the same data split results under a specific sample percentage across different distillation pairs. Our DSKD consistently outperforms Baseline and KD by 4.8% and 1.0% on average under few-shot scenarios, respectively. The traditional KD only forces the student to align class probability distributions to the teacher, leading to overfitting on the training set. By contrast, DSKD introduces diffusion model to assist feature distillation, making the student learn better feature representations and generalization capability.

### B.8. Results on Noisy-Label Scenario

The real-world scenario often has many noise labels for model training. Therefore, we further verify the effectiveness of DSKD under noisy-label scenario. As shown in Fig.9, we disturb 30%, 50%, and 70% training samples with error labels, while reserving the original test set. We found that DSKD achieves better accuracy than KD across various noise data ratios.

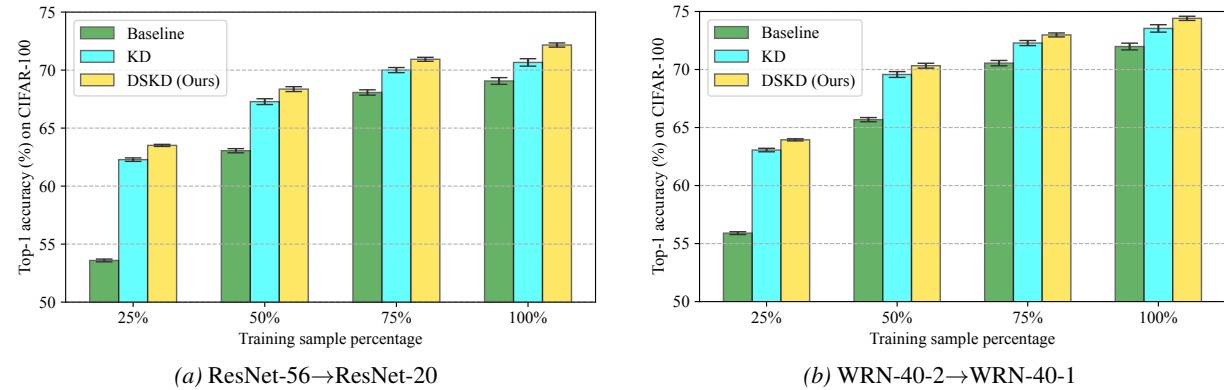

*(a)* ResNet-56→ResNet-20  *(b)* WRN-40-2→WRN-40-1

*Figure 8.* Top-1 accuracy (%) among Baseline, KD, and our DSKD under various training sample percentages.

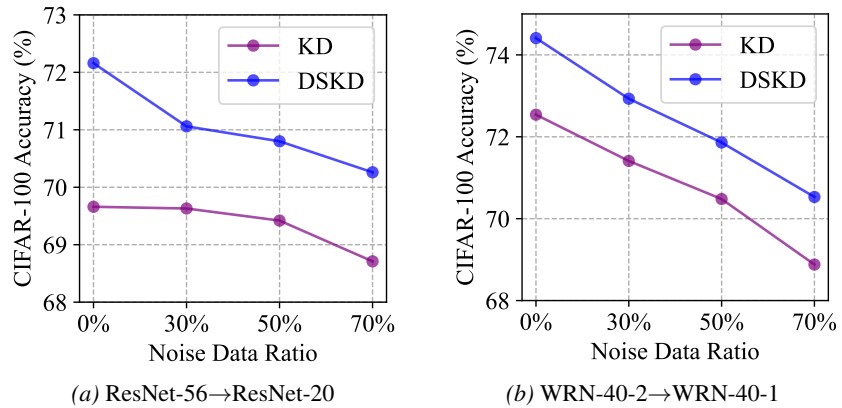

*(a)* ResNet-56→ResNet-20  *(b)* WRN-40-2→WRN-40-1

*Figure 9.* Top-1 accuracy (%) among Baseline, KD, and our DSKD under various noise data ratios.

The results demonstrate that teacher-guided diffusion distillation makes the student learn more ground-truth knowledge from the teacher than highly relying on annotated labels. Such method augments the student's defend capability to label noise.

