# OpenReview forum: "Teacher-Guided Student Self-Knowledge Distillation Using Diffusion Model"
_ICML.cc/2026/Conference — Submitted to ICML 2026_

### Official Review · Reviewer_HEhs · 2026-03-09

**Soundness:** 3
**Presentation:** 3
**Significance:** 3
**Originality:** 3
**Overall Recommendation:** 4
**Confidence:** 4

**Summary:**

The paper proposes a novel knowledge distillation framework called Teacher-Guided Student Diffusion Self-KD (DSKD). To mitigate the feature distribution discrepancy between teacher and student models , the authors introduce a light-weight diffusion model to denoise student features. Crucially, this denoising process is guided by the pre-trained teacher classifier. The resulting "denoised" student features are then used as pseudo-teachers to distill the original student features. Furthermore, the authors propose a Locality-Sensitive Hashing (LSH) guided global distillation loss to align feature directions rather than magnitudes. The method is evaluated on image classification, object detection, and semantic segmentation tasks.

**Compliance With Llm Reviewing Policy:**

Affirmed.

**Final Justification:**

Although the significant computational overhead yields only marginal empirical gains, the core concept introduces a highly innovative and valuable perspective to Knowledge Distillation. Therefore, I am inclined to accept this work.

**Key Questions For Authors:**

1. Propositions A.3 and A.4 rely strictly on the assumption that the feature vectors $v^{(stu)}$ and $\hat{v}^{(stu)}$ follow a standard normal distribution $\mathcal{N}(0, I_D)$. Given that post-activation features in architectures like ResNet are typically non-negative (e.g., post-ReLU) and decidedly non-Gaussian, how do you justify the applicability of these proofs to your actual neural network implementations?

2. In Table 8, DSKD reports lower training time and memory than DiffKD. However, DSKD requires calculating $\nabla_{x_t} \log p(y|x_t; \phi^{(tea)})$ at each of the $T$ denoising steps. Calculating these gradients w.r.t the inputs typically incurs massive computational graph retention and memory allocation overhead. Could you provide a detailed, granular breakdown of the memory and time overhead, specifically addressing how the backward passes through the teacher classifier do not explode the VRAM usage compared to a standard forward pass?

The core idea of using a teacher-classifier-guided diffusion model for KD is undeniably interesting and conceptually novel. For this reason, my initial inclination is a **Weak Accept**. However, my biggest **concerns** lie squarely in the structural invalidity of the **theoretical assumptions** (the Gaussian feature assumption) and the claims regarding **training time and memory overhead**. If these two specific and critical issues are not convincingly addressed in the rebuttal, I will definitely **decrease my score**.

**Limitations:**

While the authors include a brief, generic Impact Statement, they do not adequately discuss the technical limitations of their work. The authors must include a dedicated Limitations section discussing the sensitivity of the guidance strength hyperparameter $k$, the potential instability introduced by gradient-guided sampling, and the strict requirement of having an accurate pre-trained teacher classifier for the guidance to function properly.

**Strengths And Weaknesses:**

Pros:

1. The core motivation of using a diffusion model to bridge the representational gap in KD is conceptually intriguing. Specifically, leveraging the teacher classifier's gradients to condition the diffusion sampling process of the student features  is a highly creative approach. It provides a fresh perspective on how to transfer knowledge implicitly, distinguishing itself from standard feature alignment or prior works like DiffKD.

2. The paper is generally well-structured. The methodology, particularly the derivation of the teacher-guided denoising process , is presented logically.

3. The authors evaluate DSKD across a solid variety of tasks, including classification, detection, and segmentation, on standard benchmarks.

Cons:

1. A major vulnerability lies in the theoretical justification for the LSH-guided global distillation. Propositions A.3 and A.4 explicitly assume that both the student features $v^{(stu)}$ and the denoised features $\hat{v}^{(stu)}$ follow a standard Gaussian distribution $\mathcal{N}(0, I_D)$. In modern deep learning architectures, features before the classifier are often strictly non-negative, due to ReLU,  or heavily skewed ( LayerNorm). Applying a zero-mean, isotropic Gaussian assumption to these hidden representations is structurally invalid, which fundamentally undermines the theoretical guarantees provided in the appendix.

2. The efficiency claims presented in Table 8 are confused. The authors state that DSKD requires 21 minutes per epoch and 2472MB of memory on ImageNet, which is lower than DiffKD. However, the reverse diffusion process in DSKD uses $T=3$ timesteps on ImageNet, and Eqn. (12) requires computing the gradient of the teacher classifier with respect to the input feature: $\nabla_{x_t} \log p(y|x_t; \phi^{(tea)})$. In practice, retaining the computation graph to calculate intermediate gradients at every denoising step during the training loop demands significant VRAM and compute overhead. As researchers who frequently evaluate heavy training pipelines on modern GPU clusters, we know these gradient calculations create severe bottlenecks. The reported efficiency numbers lack a convincing explanation of how this overhead was mitigated.

3. While the idea is novel, the performance gains are marginal. For instance, on ImageNet (ResNet-34 to ResNet-18), DSKD achieves 72.57%, compared to DiffKD's 72.22%. A +0.35% improvement is difficult to justify against the added algorithmic complexity of calculating diffusion trajectories and classifier gradients.

---

> ### Author Rebuttal · Authors · 2026-03-30
>
> Thanks for your valuable comments!
>
> **Q1 and W1: Assumption of $v^{(stu)}$ and $\hat{v}^{(stu)}$ following the standard Gaussian distribution.**
>
> A1: Sorry for possible misunderstanding. First, we agree that modern neural networks often output non-negative (e.g., post-ReLU) features. However, in fact, $v^{(stu)}$ and $\hat{v}^{(stu)}$ are post-processed by a linear projection (linear layer+Z-Score Standardization) from the original non-negative student features. Here, the linear weight matrix is initialized by a zero-mean standard Gaussian distribution. According to the Central Limit Theorem, the weighted sum of a large number of independent feature dimensions (projected by the Gaussian matrix) will tend toward a Gaussian distribution. Finally, we apply Z-Score Standardization to make the distribution as a zero-mean standard Gaussian distribution. In summary, even with strictly non-negative or heavily skewed (LayerNorm) input features, the linear combination of thousands of independent Gaussian random variables will produce an approximately Gaussian distribution.
>
>
> By the way, even with minor residual skewness, the LSH mechanism is provably robust to small deviations from the strict Gaussian distribution, as established in the original LSH literature (Datar et al., 2004), which extends LSH guarantees to the entire family of p-stable distributions (Gaussian is the 2-stable case) and sub-Gaussian distributions.
>
>
> Empirically, we visualize histograms and quantile-quantile (Q-Q) plots for $v^{(stu)}$ and $\hat{v}^{(stu)}$, and verify that they conform to a standard Gaussian distribution.
>
>
> **Q2 and W2: Extra training time and memory**
>
> A2: The extra training costs introduced by the teacher classifier is relatively negligible compared to the forward and backward passes of the main backbone, due to the **highly light-weight architecture, less denoising steps, and no weight update requirement**.
>
> (1) **Highly light-weight architecture of the teacher classifier**. As described in Section 3.2, the teacher classifier $\phi^{(tea)}$ used for guidance is extremely lightweight, consisting only of a global average pooling (GAP) layer and a single linear weight matrix. Calculating the gradient $\nabla_{x_t} \log p(y | x_t; \phi^{(tea)})$ involves only a simple backpropagation through this linear head. For ImageNet-1K experiments, this linear layer only has a parameter size of 512$\times$1000 (for ResNet-34 teacher), which only accounts for 4.4% parameter size and 0.03% FLOPs.
>
> (2) **Less denoising steps**. DiffKD uses 5 denoising steps, while our DSKD only needs 3 denoising steps, due to our advanced teacher-guided diffusion model. This means the gradient calculation of the teacher classifier only needs to be performed 3 times per training iteration, with no cumulative overhead from long diffusion steps compared to DiffKD.
>
> (3) **Frozen module with no weight update**. The pre-trained teacher classifier and diffusion model remain frozen with fixed weights throughout the student training pipeline. Unlike trainable modules that require retaining the full computational graph for backward weight updates, the gradient calculation here only needs a one-time forward pass to get the class probability, followed by a lightweight gradient computation w.r.t the input feature. This process does not need to store the complete backward computational graph for parameter updates.
>
>
> **W3: Marginal gains on ResNet-34 to ResNet-18**
>
> A3: The benchmark on ImageNet (ResNet-34 to ResNet-18) is famous, thus the performance upper-bound has nearly been reached.
> We emphasize our DSKD outperforms DiffKD significantly on more challenging detection (by 0.67% AP) and segmentation (by 1.73% mIoU).
>
>
>
> **Limitation section**
>
> **L1: Sensitivity of the guidance strength**
>
> A4: The detailed sensitivity analysis has been shown in Figure 2 and Section 4.5 in paper.
> When $k>1$, the distribution becomes sharper, resulting
> in higher fidelity (but less diverse) image features. We found that $k=1$ and $k=2$ achieve the best performance on CIFAR-100 and ImageNet, respectively. Lower $k$ may weaken the strength of teacher class-aware guidance, while higher $k$ would decrease the diversity of image features.
>
> **L2: Instability of gradient-guided sampling**
>
> A5: An excessively large guidance strength $k$ can lead to "over-optimization," where features become overly sharp but lose diversity, or even drift out of the natural feature distribution. An appropriate $k$ would avoid the potential instability of gradient-guided sampling.
>
>
> **L3: An accurate teacher classifier**
>
> A6: Since the guidance signal is derived from the teacher's log-probability gradients, any inherent biases or inaccuracies in the teacher will be propagated to the student. This highlights that a good teacher is very important, which is often also a prerequisite for the traditional teacher-student distillation. As in Reviewer WFxH's Table 2, DSKD can still bring a well improvement under a weak teacher.

---

> > ### Author Rebuttal · Reviewer_HEhs · 2026-04-03
> >
> > Thanks for your detailed response. Most of my concerns have been addressed, but the additional computational cost introduced by denoising steps is unaveilable, which is a limitation of this method. Therefore, I will keep the rating as ``Weak Accept".

---

> > > ### Author Response · Authors · 2026-04-03
> > >
> > > Dear reviewer,
> > >
> > > We sincerely appreciate your time and constructive feedback throughout the review process.
> > > As shown in Table 1, we show a more detailed analysis of the additional computational cost introduced by diffusion denoising steps, and conduct a comprehensive comparison with DiffKD. The additional computational cost introduced by diffusion denoising steps with teacher guidance only **accounts for 18.7% of the total training time**.
> > >
> > >
> > > Table 1: Computational cost statistics of our DSKD and DiffKD over ResNet-18 on ImageNet run on an NVIDIA A800 GPU.
> > > | Component | DSKD | DiffKD | Key Difference & Explanation |
> > > | -- | :--: | :--: | -- |
> > > | Student network forward/backward | 17.4min | 18.6min | Almost identical, because the student backbone is exactly the same |
> > > | Diffusion model denoising | 3.7min |12.6min| DSKD uses a more lightweight diffusion backbone with fewer denoising steps, therefore achieves 70.6% lower training time than DiffKD |
> > > | Teacher classifier gradient calculation | 0.3min | - | Negligible overhead (only 1.4% of total training time), from 3 steps of linear layer gradient computation |
> > > | Total per-epoch training time |21.4min| 31.2min | DSKD achieves 31.4% lower training time than DiffKD |
> > > | Peak VRAM usage | 2472MB |  3275MB | DSKD has 24.5% lower VRAM usage than DiffKD |
> > >
> > > Your valuable comments have significantly strengthened our paper, and we will follow your suggestions to further improve our work. Given the rapid development of the field of diffusion models, we hope our basic framework will inspire future works to combine visual distillation with more advanced and well-designed diffusion models to pursue better performance with less training time.
> > >
> > > Best regards,
> > >
> > > Authors

---

### Official Review · Reviewer_AP9N · 2026-03-09

**Soundness:** 3
**Presentation:** 3
**Significance:** 3
**Originality:** 3
**Overall Recommendation:** 4
**Confidence:** 2

**Summary:**

This paper studies the difficulty of knowledge distillation caused by the mismatch between teacher and student feature distributions, and proposes a diffusion-based teacher-guided student self-distillation framework, termed DSKD. Instead of directly aligning intermediate features between teacher and student, the method first trains a lightweight diffusion model using teacher features, and then denoises student features under the guidance of the teacher classifier to obtain denoised student features infused with the teacher’s discriminative information, which are used as the distillation target for the original student features. The paper further designs a local MSE-based distillation loss and an LSH-based global distillation loss to emphasize semantic directional consistency. Experiments cover multiple vision tasks, including image classification, object detection, and semantic segmentation, and the method achieves competitive results on both homogeneous and heterogeneous architectures. Overall, the main contributions of this work are: introducing teacher-classifier-guided diffusion sampling into feature distillation, proposing a self-distillation paradigm that distills original student features from denoised student features to mitigate the teacher-student representation gap, and designing a distillation objective that combines local and global constraints.

**Compliance With Llm Reviewing Policy:**

Affirmed.

**Final Justification:**

I am maintaining my current score.

**Key Questions For Authors:**

1. Regarding the role of the teacher classifier in Section 3.2, please clarify the distinction between the training stage and the sampling stage. The description in Algorithm 1 and the methodology in Section 3.2 feel somewhat disconnected, especially concerning the role of the teacher classifier. Please explain this in more detail.

2. The method figure is missing the local and global distillation process between the original student features and the denoised student features, and it also does not show the loss terms. A more complete figure would significantly improve the readability of the paper.

3. The paper argues that LSH focuses more on feature direction than magnitude, but cosine similarity or angular losses have similar properties. Why did the authors choose the random projection + BCE-based hashing formulation instead of a more direct directional alignment loss? The distinction among these design choices should be made clearer.

4. The paper describes the method as student self-distillation, but the denoised target is explicitly obtained using a diffusion model trained on teacher features and guidance from the teacher classifier. Could the authors further clarify the boundary of the term “self-distillation” here? Also, if other self-distillation methods were equipped with similar teacher guidance, would DSKD still maintain its advantage?

**Limitations:**

The paper provides fairly thorough analysis of performance, efficiency, and experimental results, but lacks sufficient discussion of limitations and potential negative impacts. I would suggest adding discussion on the following aspects:

1. he dependence of the method on the quality of the teacher classifier;
2. the risk that guidance may fail or even mislead the student under noisy labels, weak teachers, or long-tailed categories;
3. the additional training complexity and deployment cost introduced by extra diffusion sampling steps. If the authors could discuss these aspects, the paper would be more complete.

**Strengths And Weaknesses:**

Strengths

This paper is overall clearly written, and the motivation is fairly natural. The authors identify a real issue in conventional feature distillation, namely the incompatibility between teacher and student feature spaces, and accordingly propose to first train a diffusion model on teacher features, then denoise student features under teacher guidance, and finally reformulate the supervision target as a student-to-denoised-student distillation objective. This idea is reasonable and also relatively clearly distinguished from DiffKD. In terms of method design, the combination of teacher-guided diffusion and self-distillation has some novelty, and the LSH-guided global distillation is not merely a repetition of existing MSE-based alignment. Experimentally, the paper covers multiple tasks, includes a variety of teacher-student architectures and baselines, and provides relatively complete ablations and parameter analyses. The empirical performance is strong, and the lightweight design also improves practical value. Overall, this work provides a useful contribution to the literature on visual distillation.

Weaknesses

1. The method figure and training pipeline are not sufficiently complete. The current framework figure does not clearly present the distillation process between the original student features and the teacher-guided denoised student features, nor does it clearly show where the different loss terms are applied in the pipeline, which makes the method somewhat harder to follow.
2. The role of the teacher classifier in Section 3.2 is not explained intuitively enough. According to Algorithm 1, it seems that the diffusion model is first trained using teacher features, and then student features are fed into the trained diffusion model for denoising. However, Section 3.2 additionally introduces teacher-classifier guidance. It is therefore unclear what exactly the teacher classifier is, what role it plays, and whether it serves as a conditional input to the diffusion model. The textual explanation in Section 3.2 is somewhat complicated, and the authors should more clearly distinguish the roles of diffusion model training and student feature sampling.
3. In Section 3.2, the approximation from the conditional distribution to the shifted-mean form, as well as the first-order Taylor expansion and low-curvature assumption between Eq.(7) and Eq.(12), are explained too quickly. More intuitive explanation would be helpful.
4. The derivation of the reverse process is somewhat too concise. The paper would benefit from including more intermediate steps as well as more details for the derivation of the loss functions.
5. The search range of the hyperparameter α\alphaα is very large, but the rationale is insufficiently explained. The authors should clarify why such a wide search range was chosen and whether this weight is related to the numerical scales of the different loss terms.
6. Why the denoised student features are necessarily better supervision targets is not fully justified. While this claim is partially supported by visualization and improved performance, the paper does not further quantify how much closer these features are to teacher features at the distribution level, nor does it show under what circumstances denoising may remove useful information originally present in the student features.

---

> ### Author Rebuttal · Authors · 2026-03-30
>
> Thanks for your valuable comments!
>
> **Q1 and W2: The teacher classifier (TeaCLS) role**
>
> A1: The TeaCLS refers to the pre-trained and frozen classification head of the teacher. The TeaCLS role is to provide class-aware guidance for denoising without being used for diffusion model training. The distinction between training and sampling: (1) During training, the diffusion model is trained on teacher features. (2) During sampling, the TeaCLS calculates the gradient that is used to shift the denoising trajectory towards teacher. This ensures denoised student features have teacher-level quality and discriminability.
> This process is Eq.(12): $x_{t-1}\sim \mathcal{N}({\mu}+k{\Sigma}\bigtriangledown_{{x}_t}\log p(y|{x}_t;{\phi}^{(tea)}), {\Sigma})$. The TeaCLS is not a direct conditional input to the diffusion model, but an external guidance perturbing ${\mu}$ in denoising.
>
> For Algorithm 1 and Section 3.2 disconnected, Section 3.2 corresponds to Lines 4-8 of Algorithm 1; the TeaCLS only acts in Line 7 (decoupled from Line 3’s diffusion training).
>
> **Q2 and W1: The figure details**
>
> A2: In figure 1, we will add "local and global distillation" details, and tag $\mathcal{L} _\mathrm{Task}$, $\mathcal{L} _\mathrm{DSKD}$, $\mathcal{L} _\mathrm{Diff}$, and $\mathcal{L} _\mathrm{KD}$ on the training pipeline.
>
>
> **W3: Shifted-mean, Taylor expansion and low-curvature assumption**
>
> A3: In fact, the details are in Appendix Section A.1. Here, we highlight critical points.
>
> (1) **Shifted-mean form**. The details are in Eq.(22). The core is:
> $$
> 	\log(p_{{\theta}}(x_{t}|{x}_{t+1}) p(y|{x}_t;{\phi}^{(tea)}))=-\frac{1}{2}({x}_t-{\mu}-{\Sigma}{g})^\top{\Sigma}^{-1}({x}_t-{\mu}-{\Sigma}{g})+C_3,
> $$
> then derives $\mathcal{N}({\mu}+{\Sigma}{g},{\Sigma})$.
>
> (2) **First-Order Taylor Expansion** is formulated as:
>
> $$
> 	\log p(y|x_t;\phi^{(tea)}) \approx \log p(y|x_t;\phi^{(tea)})|_{x_t=\mu}+(x_t-\mu)\bigtriangledown _{x_t}\log p(y|x_t;\phi^{(tea)})| _{x_t=\mu}
> 	=(x_t-\mu)g+C_1,
> $$
>
> this guides the model to shift features for better matching class $y$ by linear guidance.
>
> (3) **Low-curvature assumption**. With a large number of denoising steps, the variance $\Sigma$ of the sampling distribution shrinks to nearly 0, making $\Sigma^{-1}$ extremely large. The curvature of $\log p(y|x_t;\phi^{(tea)})$ is negligible in this local range, meaning the log probability can be safely approximated as a linear function around $\mu$.
>
> **W4: More intermediate steps and details**
>
> A4: The full steps can be found in Appendix Section A.1.
> Derivation of loss functions is:
>
> $$\nabla_{f^{(\text{stu})}} \mathcal{L}_{\text{DSKD}} = 2 \left( f^{(\text{stu})} - \hat{f}^{(\text{stu})} \right) - \frac{\gamma}{M} \sum _{m=1}^{M} (\delta_m - \rho_m) \cdot \frac{W_m}{\| f^{(\text{stu})} \|_2}.$$
>
> **W5: Search range of $\alpha$.**
>
> A5: The fine-grained range of $\alpha$ is in Table 1, and $\alpha=1$ achieves the best, increasing $\alpha$ degrades accuracy. $\alpha$ is only related to the numerical scales of $\mathcal{L}_\mathrm{DSKD}$ (no impact on other losses), which is empirically verified.
>
> Table 1: impact on $\alpha$
> | $\alpha$ | 0.5 | 1.0 | 2.0 |5.0 | 10.0 | 20.0 | 50.0 | 100.0 |
> | -- | :--: | :--: |:--: |:--: | :--: | :--: |:--: |:--: |
> | Acc | 73.96 | **74.45** | 74.34 | 74.35 |74.28 | 73.92| 73.75 | 73.72|
>
>
> **W6: how much closer to teacher and when remove useful information**
>
> A6: Table 2 quantifies student-teacher feature distribution via FID/KL: denoising reduces FID (1923.99→999.22) and KL (11.59→2.48), indicating closer to teacher features. Denoising may remove useful information only if the diffusion model is poorly trained (avoided in our work due to well-trained models).
>
> Table 2: FID and KL
> | Metric | Before denoising | After denoising |
> | :--: | :--: | :--: |
> | FID | 1923.99 | **999.22** |
> | KL | 11.59 | **2.48** |
>
>
> **Q3: LSH v.s. cosine/angular**
>
> A7: Cosine/angular losses constrain feature direction consistency via a single global scalar (failing to capture fine-grained semantic subspace distribution differences). Our LSH uses $M$ hash functions to partition high-dimensional space into $M$ orthogonal semantic subspaces, forcing student features to align with denoised features in each subspace (not just global direction).
> In Table 3, LSH outperforms cosine and angular losses by 0.37\% and 0.49\%.
>
> Table 3: LSH v.s. cosine/angular
> | Cosine | Angular | LSH |
> | :--: | :--: | :--: |
> | 74.08 | 73.96 | **74.45** |
>
> **Q4: Clarify “self-distillation”**
>
> A8: Here, “self-distillation” indicates we use denoised student features to distill the student itself. We agree the denoised target is learn from the teacher, and we will remove “self-distillation” terms to avoid possible confusing. In Table 4, DSKD also achieves the best.
>
> Table 4: Accuracy of self-distillation+teacher on ImageNet
> | MixSKD | FASD | DSKD (Ours) |
> | :--: | :--: | :--: |
> | 72.04 | 71.86 | **72.57** |
>
> **Limitations** Please refer to Reviewer WFxH's Q1 and Reviewer HEhs's Q2 and Limitations.

---

> > ### Author Rebuttal · Reviewer_AP9N · 2026-04-02
> >
> > Thank you for the detailed responses. The clarifications addressed some of my concerns. I am maintaining my current score.

---

> > > ### Author Response · Authors · 2026-04-02
> > >
> > > Dear reviewer,
> > >
> > > We sincerely appreciate your time and constructive feedback throughout the review process. Due to the limited space during the rebuttal stage, we show more detailed theoretical steps here.
> > >
> > > **W3: More intuitive explanation of the approximation from the conditional distribution to the shifted-mean form, as well as the first-order Taylor expansion and low-curvature assumption between Eq.(7) and Eq.(12).**
> > >
> > > A3: We apologize for the brief explanation of this part. In fact, the detailed explanation and theoretical analysis are supplied in Appendix Section "A.1. Teacher-Guided Student Feature Denoising". Here, we re-highlight several critical points to provide more intuitive explanation.
> > >
> > > (1) **Approximation from the conditional distribution to the shifted-mean form**. The transition from Eq.(7) to Eq.(12) follows the logic of "completing the square" for the product of two exponential functions. When we multiply the unconditional Gaussian (Eq.(9)) by the linearized classifier term (Eq.(10)), the resulting log-distribution remains quadratic:
> > >
> > > $$
> > > \begin{align*}
> > > 	&\log(p_{{\theta}}(x_{t}|{x}_{t+1}) p(y|{x}_t;{\phi}^{(tea)}))  \\\\
> > > 	&\approx  -\frac{1}{2}({x}_t-{\mu})^\top{\Sigma}^{-1}({x}_t-{\mu})+({x}_t-{\mu}){g}+C_2  \\\\
> > > 	&=-\frac{1}{2}({x}_t-{\mu}-{\Sigma}{g})^\top{\Sigma}^{-1}({x}_t-{\mu}-{\Sigma}{g})+\frac{1}{2}{g}^\top{\Sigma}{g}+C_2  \\\\
> > > 	&=-\frac{1}{2}({x}_t-{\mu}-{\Sigma}{g})^\top{\Sigma}^{-1}({x}_t-{\mu}-{\Sigma}{g})+C_3  \\\\
> > > 	&=\log p({z})+C_4.
> > > \end{align*}
> > > $$
> > >
> > > Therefore, we can derive ${z}\sim \mathcal{N}({\mu}+{\Sigma}{g},{\Sigma})$.
> > >
> > > Due to $g=\bigtriangledown_{x_t}\log p(y|x_t;{\phi}^{(tea)})|_{{x}_t={\mu}}$,
> > >
> > > the teacher-guided diffusion denoising is formulated as:
> > > $x_{t-1}\sim \mathcal{N}({\mu}+k{\Sigma}\bigtriangledown_{{x}_t}\log p(y|{x}_t;{\phi}^{(tea)}), {\Sigma}).$
> > >
> > > (2) **Intuition for the First-Order Taylor Expansion**. The approximation of $\log p(y|{x}_t;{\phi}^{(tea)})$ by a first-order Taylor expansion at ${x}_t={\mu}$ is formulated as:
> > >
> > > $$
> > > \begin{align*}
> > > 	\log p(y|x_t;\phi^{(tea)}) & \approx \log p(y|x_t;\phi^{(tea)})|_{x_t=\mu}+(x_t-\mu)\bigtriangledown _{x_t}\log p(y|x_t;\phi^{(tea)})| _{x_t=\mu}  \\\\
> > > 	&=(x_t-\mu)g+C_1,
> > > \end{align*}
> > > $$
> > >
> > > The first-order Taylor expansion in Eq.(10) serves to linearize the classifier's "guidance". By approximating $\log p(y | x_t; \phi^{(tea)})$ as a linear function $(x_t - \mu)g + C_1$, we essentially treat the classifier as providing a constant directional "force" or "nudge" $g$ that tells the model which way to shift the feature to better match class $y$. This simplifies a complex, non-linear probability landscape into a simple local gradient descent step.
> > >
> > > (3) **Rationality of the low-curvature assumption**. This is not an arbitrary artificial constraint, but a natural property of the multi-step diffusion denoising process. For diffusion models with a sufficiently large number of denoising steps, the variance $\Sigma$ of the Gaussian sampling distribution in each single step shrinks to nearly 0, making the inverse covariance $\Sigma^{-1}$ extremely large. In comparison, the curvature (second-order derivative) of the teacher classifier’s log probability $\log p(y|x_t;\phi^{(tea)})$ is negligible in this local range, meaning the log probability can be safely approximated as a linear function around the diffusion model’s predicted mean $\mu$.
> > >
> > > **W4: More intermediate steps and details for the derivation.**
> > >
> > > A4: We will supplement complete intermediate derivation steps.
> > >
> > > **The derivation of the reverse process.**
> > > we will expand Appendix A.1 to provide a more exhaustive step-by-step derivation. Specifically, starting from the log-joint distribution in Eq.(7):
> > >
> > > $$\log p(x_t | x_{t+1}, y) = \log p_\theta(x_t | x_{t+1}) + \log p(y | x_t; \phi^{(tea)}) + C.$$
> > >
> > > Using the Gaussian form for $p_\theta$ and the first-order Taylor expansion for the classifier log-probability at $x_t = \mu$, we obtain:
> > >
> > > $$\approx -\frac{1}{2}(x_t - \mu)^\top \Sigma^{-1} (x_t - \mu) + (x_t - \mu)g + C_2.$$
> > >
> > > To complete the square for $x_t$, we rearrange the terms as:
> > >
> > > $$= -\frac{1}{2} [ (x_t - \mu)^\top \Sigma^{-1} (x_t - \mu) - 2(x_t - \mu)^\top \Sigma^{-1} (\Sigma g) + g^\top \Sigma g - g^\top \Sigma g ] + C_2$$
> > >
> > > $$= -\frac{1}{2} (x_t - \mu - \Sigma g)^\top \Sigma^{-1} (x_t - \mu - \Sigma g) + \frac{1}{2}g^\top \Sigma g + C_2.$$
> > >
> > > Since $\frac{1}{2}g^\top \Sigma g$ is independent of $x_t$, it is absorbed into the constant. This directly yields the shifted Gaussian $\mathcal{N}(\mu + \Sigma g, \Sigma)$, which is the foundation for our guided sampling Eq.(12).
> > >
> > >
> > > Your valuable comments have significantly strengthened our paper, and we will follow your suggestions to further improve our work.
> > >
> > > Best regards,
> > >
> > > Authors

---

### Official Review · Reviewer_WFxH · 2026-03-11

**Soundness:** 3
**Presentation:** 3
**Significance:** 2
**Originality:** 3
**Overall Recommendation:** 3
**Confidence:** 3

**Summary:**

The paper introduces DSKD (Teacher-Guided Student Diffusion Self-KD), a novel knowledge distillation method that uses a teacher-classifier-guided diffusion model to denoise student features and perform self-distillation. By addressing the feature distribution discrepancies between teacher and student models, DSKD achieves state-of-the-art performance in classification, detection, and segmentation tasks.

**Compliance With Llm Reviewing Policy:**

Affirmed.

**Final Justification:**

I am maintaining my current score.

**Key Questions For Authors:**

What are the potential challenges in deploying DSKD in real-world applications with limited computational resources?
How sensitive is the method to hyperparameters like the number of diffusion steps or LSH hash functions?
Why were SimKD and ReviewKD not included in the experimental comparisons (if omitted)? Are there specific reasons for their exclusion?

**Limitations:**

The method may require additional fine-tuning of hyperparameters (e.g., diffusion steps, hash function count), which could hinder ease of use.

**Strengths And Weaknesses:**

Strengths
Innovative Approach: The use of a teacher-classifier-guided diffusion model for denoising student features is novel and effectively addresses the teacher-student feature mismatch problem.
Comprehensive Results: Demonstrates significant performance improvements across various tasks and datasets, including classification, detection, and segmentation.

Weaknesses
The method has limited discussion on scalability for extremely large-scale datasets or real-time applications.
The use of diffusion models and LSH-guided loss increases implementation complexity compared to simpler distillation methods.

---

> ### Author Rebuttal · Authors · 2026-03-30
>
> Thanks for your valuable comments!
>
> **W1: On extremely large-scale datasets.**
>
> A1: We conduct experiments on ScanNetV2, a large-scale indoor 3D scene dataset for 3D semantic segmentation. The dataset is approximately 1.5TB in total size and contains over 2.5 million RGB-D frames. In Table 1, The distillation experiments are conducted on Point Transformer V3 (PTv3) [1]. Our DSKD improve the student baseline by 3.56\% on PTv3, demonstrating the good scalability.
>
> Ref: [1] Point Transformer V3: Simpler, Faster, Stronger. CVPR 2024.
>
>
> Table 1: PTv3 on ScanNetV2.
>
> | Method        | Params (M) | FLOPs (G) | mIoU (%) |
> |---------------|------------|-----------|----------|
> | Teacher: PTv3       | 46.17M     | 641.24G   | 77.31    |
> | Student: PTv3-Small     |     2.78M       |      167.10G     | 72.37    |
> | +KD           |            |           | 73.06    |
> | +PVKD (CVPR 2022)          |       |    | 74.69    |
> | +TGKD (arXiv:2505)          |            |           | 74.66    |
> | +DSKD (ours)  |            |           | **75.93**|
>
>
>
>
> **W2: Real-time applications.**
>
> A2: DSKD is a training-only distillation framework, it introduces zero additional overhead during inference, supporting real-time applications. In Table 2 of our paper, our DSKD can improve the lightweight MobileNetV1 (widely used in real-time edge applications) by 3.6\% accuracy, proving strong scalability to real-time deployment scenarios.
>
>
> **W3: Implementation complexity.**
>
> A3: The diffusion models and LSH-guided loss can be easily implemented by standard operators in modern deep learning frameworks like PyTorch. We will release the full, well-annotated PyTorch implementation code, including pre-trained diffusion model weights, a one-click reproduction script. This will allow other researchers to quickly apply our method to their own tasks with minimal development cost.
>
> **Q1: Potential challenges in deploying DSKD.**
>
> A4: If DSKD is deployed on edge devices for real-world applications with limited computational resources, the main challenge is the iterative nature of the diffusion process. Even though we have optimized the process to only 2 or 3 steps, performing multiple forward passes of the diffusion backbone still increases latency compared to a single-pass backbone.
>
> In Table 2, under noisy labels, weak teachers, or long-tailed categories, we find DSKD is more robust than Baseline (without distillation) and KD, and also achieves remarkable performance gains.
>
> Table 2: Accuracy on CIFAR-100
> | Condition | Baseline | KD | DSKD |
> | -- | :--: | :--: | :--: |
> | 50% noisy label | 56.35% | 62.45%  | **70.16%** |
> | weak teacher (Acc=62%) | 71.98% | 71.33% | **73.42%** |
> | long-tailed categories (10% tailed samples) | 58.87% | 61.56% | **64.79%** |
>
>
> **Q2: The number of diffusion steps or LSH hash functions.**
>
> A5: The experiments of the number of diffusion steps and LSH hash functions have shown in Table 10 and 11 of our paper, respectively. For ease of display, we place tables as follows:
>
> Table 3: Accuracy of various diffusion step numbers
>
> |Timesteps $T$ | 1 | 2 | 3 | 4 | 5 |
> | :--: | :--: | :--: | :--: | :--: | :--: |
> | CIFAR-100| 74.20 | 74.45 |74.42 | 74.49 | 74.47 |
> |ImageNet| 71.93 | 72.34 | 72.57 | 72.58 | 72.60 |
>
> Table 4: Accuracy of various LSH hash function numbers
>
> | Hash function number $M$ | 32 | 64 | 128 | 256 | 512 |
> | :--: | :--: | :--: | :--: | :--: | :--: |
> | CIFAR-100 | 74.28 | 74.36 | 74.39 | **74.45** | 74.44 |
> | ImageNet | 72.38 | 72.49 |72.51 | **72.57** | 72.55 |
>
>
> we found that $T=2$ on CIFAR-100 and $T=3$ on ImageNet are good enough for DSKD for pursuing the best efficiency-accuracy trade-off. We observe that $M=256$ achieves the best performance. Overall, DSKD is not sensitive to the number of diffusion steps or LSH hash functions, and produces consistent performance gains.
>
>
> **Q3: Comparison with SimKD and ReviewKD.**
>
> A6: In Table 5, our DSKD outperforms SimKD and ReviewKD by an average improvement of 1.10\% and 1.07\% on ImageNet, respectively. We will add this comparison to our paper.
>
>
> Table 5: Accuracy of various distillation methods on ImageNet.
>
> | Teacher | ResNet-34 | ResNet-50 |
> | :--: | :--: | :--: |
> | Student | ResNet-18 | MobileNetV1 |
> | Baseline | 69.76 | 70.13 |
> | SimKD | 71.66 | 72.44 |
> | ReviewKD | 71.61 | 72.56 |
> | DSKD (Ours) | **72.57** | **73.73** |

---

> > ### Author Rebuttal · Reviewer_WFxH · 2026-04-07
> >
> > Q3: Comparison with SimKD and ReviewKD
> >
> > The results reported for SimKD on WRN-40-2 → WRN-40-1 and ResNet-32x4 → ResNet-8x4 are 75.56 and 78.08, respectively, which are both higher than your reported results of 74.45 and 77.08. Therefore, I do not agree with the claim that your method is effective across all common distillation pairs.
> >
> > I will keep my original score.

---

> > > ### Author Response · Authors · 2026-04-08
> > >
> > > Dear reviewer,
> > >
> > > We sincerely appreciate your time and constructive feedback throughout the review process. As for the comparison with SimKD on WRN-40-2 → WRN-40-1 and ResNet-32x4 → ResNet-8x4, we found that **the original SimKD literature adopts teacher or student with higher accuracy than ours to conduct distillation experiments**. For a fair comparison, we reproduced DSKD under the same teacher checkpoint and training setups with the SimKD official code, as shown in Table 1. **DSKD can still achieve better accuracy than SimKD by improvements of 0.41% and 0.37% on WRN-40-1 and ResNet-8x4**, respectively. The results demonstrate that DSKD is better than SimKD under the exactly same training conditions.
> > >
> > > Table 1: Accuracy of various distillation methods on CIFAR-100.
> > >
> > > | Teacher | WRN-40-2 | ResNet-32x4 |
> > > | -- | :--: | :--: |
> > > | Teacher Accuracy | 76.31 | 79.42 |
> > > | Student | WRN-40-1 | ResNet-8x4 |
> > > | Baseline | 71.92 ± 0.17 | 73.09 ± 0.30 |
> > > | SimKD | 75.56 ± 0.27 | 78.08 ± 0.15 |
> > > | DSKD (Ours) | **75.97** ± 0.24 | **78.45** ± 0.21|
> > >
> > > Moreover, we emphasize that CIFAR-100 is a relatively small-scale dataset in the era of modern deep learning. Its performance gain nearly **reaches the upper bound**, which may limit its effectiveness for distinguishing between high-performing distillation methods. In contrast, we highlight the experiments on the large-scale ImageNet, as shown in rebuttal (also shown below for convenience), **our DSKD outperforms SimKD and ReviewKD by an average improvement of 1.10\% and 1.07\% on ImageNet**, respectively. The more convincing results on the large-scale ImageNet verify that our DSKD is a better method than SimKD.
> > >
> > >
> > > Table 2: Accuracy of various distillation methods on ImageNet.
> > >
> > > | Teacher | ResNet-34 | ResNet-50 |
> > > | -- | :--: | :--: |
> > > | Student | ResNet-18 | MobileNetV1 |
> > > | Baseline | 69.76 | 70.13 |
> > > | SimKD | 71.66 | 72.44 |
> > > | ReviewKD | 71.61 | 72.56 |
> > > | DSKD (Ours) | **72.57** | **73.73** |
> > >
> > >
> > > Your valuable comments have significantly strengthened our paper, and we will include this comparison in the final version to further improve our work.
> > >
> > > Best regards,
> > >
> > > Authors

---

### Decision · Program_Chairs · 2026-04-30

**Decision:**

Reject

**Comment:**

Reviewers agreed that the core idea — using teacher-classifier-guided diffusion sampling to bridge the teacher-student feature distribution gap — is conceptually novel and clearly distinguished from prior work such as DiffKD. The evaluation spans classification, detection, and segmentation tasks with ablation studies. The rebuttal was substantive, providing efficiency breakdowns, distribution-level measurements (FID/KL), and large-scale experiments (ScanNetV2) that addressed the majority of reviewer concerns. Two of three reviewers recommend Weak Accept; the third maintains Weak Reject primarily over a disputed baseline comparison.

Considering all reviews, rebuttal, and discussions, the paper currently falls on the borderline, but a major revision is required. First, the substantive rebuttal with many new datasets, baselines, and results triggers the need for a major revision. Second, the performance gains on standard ImageNet benchmarks are modest relative to the added training complexity, which remains a known limitation of the method. Third, the following issues must be addressed through a major revision: (1) The comparison with SimKD and ReviewKD under identical training conditions — demonstrated in the rebuttal to favor DSKD — must be incorporated into the main paper, as the original submission's numbers on CIFAR-100 were not directly comparable due to differing training setups. (2) The method figure should be updated to annotate all loss terms and clearly illustrate the local/global distillation process between original and denoised student features. (3) The term "self-distillation" should be removed or reframed, as the denoised target is explicitly conditioned on teacher knowledge, which may mislead readers. (4) A dedicated limitations section should be added covering the sensitivity of the guidance strength hyperparameter α, the dependence on a reliable teacher classifier, and the unavoidable training overhead introduced by diffusion denoising steps. (5) The FID/KL distribution measurements and the efficiency breakdown table provided in the rebuttal should be incorporated into the main paper to substantiate the core claims.